# Semaphorin heterodimerization in *cis* regulates membrane targeting and neocortical wiring

Paraskevi Bessa [1,9], Andrew G. Newman [1,9], Kuo Yan[1], Theres Schaub[1], Rike Dannenberg[1], Denis Lajkó[1], Julia Eilenberger[2], Theresa Brunet [3,4], Kathrin Textoris-Taube[5,6], Emanuel Kemmler [7], Penghui Deng[1], Priyanka Banerjee[7], Ethiraj Ravindran[1], Robert Preissner [7], Marta Rosário [1,10] & Victor Tarabykin [1,8,10] ✉

Disruption of neocortical circuitry and architecture in humans causes numerous neurodevelopmental disorders. Neocortical cytoarchitecture is orchestrated by various transcription factors such as Satb2 that control target genes during strict time windows. In humans, mutations of SATB2 cause SATB2 Associated Syndrome (SAS), a multisymptomatic syndrome involving epilepsy, intellectual disability, speech delay, and craniofacial defects. Here we show that Satb2 controls neuronal migration and callosal axonal outgrowth during murine neocortical development by inducing the expression of the GPI-anchored protein, Semaphorin 7A (Sema7A). We find that Sema7A exerts this biological activity by heterodimerizing in cis with the transmembrane semaphorin, Sema4D. We could also observe that heterodimerization with Sema7A promotes targeting of Sema4D to the plasma membrane in vitro. Finally, we report an epilepsy-associated de novo mutation in Sema4D (Q497P) that inhibits normal glycosylation and plasma membrane localization of Sema4D-associated complexes. These results suggest that neuronal use of semaphorins during neocortical development is heteromeric, and a greater signaling complexity exists than was previously thought.

Accurate axonal outgrowth, navigation, and neuronal migration are critical for the establishment of functional neocortical circuits. Defects in axonal outgrowth or neuronal migration are frequently associated with cognitive impairment as well as with severe epilepsy[1,2]. After their birth, neurons undergo a polarization process in the subventricular zone (SVZ) and intermediate zone (IZ) of the developing neocortex, which is essential both for specification and initial extension of axons and for initiation of migration into the cortical plate[3,4]. Polarization and the ensuing radial migration are regulated both by cell-intrinsic factors (cell autonomous) as well as by interaction with environmental cues (non-cell autonomous)[5].

In the last two decades, several transcription factors have been identified, whose ablation causes abnormal development of projection neurons in the neocortex (see reviews from refs. 6–9). Deletion of the *Satb2 gene* (Special AT-rich sequence binding 2) interferes with radial migration, fate determination and axonal projection of late-born neurons, resulting in agenesis of the corpus callosum[10,11]. In humans, mutations of the SATB2 gene cause SATB2 Associated Syndrome (SAS) and are characterized by symptoms such as developmental delay (DD)/ intellectual disability (ID), epilepsy, absent or limited speech development, craniofacial abnormalities including palatal and dental abnormalities, dysmorphic features and behavior[12–15].

The Semaphorin family is a large family of diverse guidance molecules that respond to environmental cues to direct multiple aspects of cellular behavior in the development of the nervous system as well as in immune cell function. Originally described as having fundamental roles in growth cone collapse and axon fasciculation[16–18], Semaphorins have since been described to regulate many other processes during the development of the nervous system, including dendritic arborization and maturation, cell sorting, cell polarization, neuronal cell migration and laminar positioning[19,20]. Mammalian Semaphorins are divided into five subfamilies consisting of secreted and membrane-bound proteins[21–23]. Semaphorin7A (Sema7A) is the only Semaphorin that is attached to the cell membrane via a C-terminal Glyco-Phosphatidyl-Inositol (GPI) anchor and lacks a cytoplasmic domain. Sema7A forms homodimers through SEMA and IG domain interactions and dimerization seems to be important for the Semaphorin function in different cell systems[24,25].

Classically, Semaphorins have been observed to act as ligands to stimulate signal transduction by binding in trans to their Plexin or Neuropilin receptors[17,26,27]. During the past few years, evidence is mounting for the existence of reverse Semaphorin signaling, whereby ligand- binding activity induces signal transduction in the Sema-containing cell. Reverse signaling has been reported for Sema6A[28], Sema6B[29], Sema6D[30], and Sema4A[31], with important implications for neuronal development and axonal pathfinding[32]. Semaphorins that can signal in reverse all have a cytoplasmic domain.

Here, we identify the GPI-linked Sema7A as a Satb2 direct target, that cell autonomously controls both radial migration and axon outgrowth of layer II-III neocortical neurons. Additionally, we report that Sema7A initiates reverse signaling by heterodimerizing with the transmembrane Semaphorin, Sema4D. Sema7A:Sema4D interaction is required for the membrane localization of Sema4D and for both the migration and axonal outgrowth of upper layer (UL) neocortical neurons. Moreover, we discovered the de novo mutation SEMA4D-Q497P in a patient with epilepsy. This mutation interferes with the targeting of Sema4D:Sema7A heterodimers to the plasma membrane and growth cones by modifying the post-translational glycosylation of SEMA4D. Furthermore, the SEMA4D-497P mutation inhibits migration and axon projections of cortical neurons in the murine-developing neocortex. Our results emphasize the importance of promoting the membrane localization of Sema4D and demonstrate the role of Sema7A and residue Q497 play in this process. Moreover, our data reveal an alternative mechanism where traditionally ligand-considered molecules, such as Sema4D and Sema7A, form heterodimeric signaling complexes to initiate reverse signaling that regulates core processes of neocortical circuit formation. This work further contributes to our understanding of the complex etiology of neurodevelopmental pathologies.

## Results

### Satb2 controls neuronal migration and corpus callosum development cell autonomously

One of the most consistent features of Satb2 Associated Syndrome (SAS), and the Satb2 knockout mouse is agenesis of the corpus callosum[11,13]. We previously found that layer II-III and layer V neurons use distinct molecular programs downstream of Satb2 to project axons to form this trans-hemispheric axonal tract[33]. Satb2 distinctly orchestrates both cell-autonomous and non-cell-autonomous transcriptional programs important for this developmental process.

In order to dissect whether Satb2 is required cell-autonomously for axon development of late-born projection neurons, we used a targeted *Satb2* mouse strain where exon 2 of Satb2 is "floxed" (*Satb2fl/fl*) and is then deleted by cre-mediated recombination. When *Satb2* is deleted from the developing dorsal neocortex using *NexCre*, we observe that the corpus callosum is not formed and some axons reroute via the internal capsule (Figure S1b), similar to what is observed in the constitutive mutant[33]. Re-expression of Satb2 in a selected number of

neurons in the Satb2-deficient cortex is not sufficient to restore projections to the corpus callosum. This implies that there are non-cell autonomous (environmental) factors in a Satb2-deficient cortex that inhibit corpus callosum formation and confound possible understanding of the cell-autonomous role of Satb2 in axon projection. When Satb2 is deleted in a mosaic fashion by only introducing Cre into only a few cells of the wild-type cortex, neuronal migration is perturbed, and axons do not project at all. Re-expression of Satb2 in the same cre-expressing cell restores axon outgrowth in that cell, demonstrating that axon outgrowth per se is controlled cell autonomously by Satb2 (Figure S1C–E).

Live imaging of organotypic slices with cell autonomous deletion of Satb2 revealed distinct differences in the behavior and morphology of neurons in the intermediate zone (IZ) compared to wildtype (Fig. S2). While wild-type neurons start migrating radially after acquiring a single leading process, *Satb2*-deficient neurons fail to leave the IZ and often formed bifurcated leading processes.

Given that cell-autonomous neuronal migration and axon extension can be fully restored when re-introducing Satb2, while non-cell autonomous effects cannot (Figure S1), we sought to understand which Satb2 targets contribute to cell-autonomous development of the corpus callosum.

### Sema7A acts downstream of Satb2 to control radial migration and axon elongation

In order to identify Satb2 downstream targets that control radial migration and axonal growth, we performed an in situ hybridization (ISH) screen with molecules known to be involved in axon formation and guidance (reviewed in refs. 34–36). In this screen we used the *NexCre* mouse strain in order to delete Satb2 only in pyramidal neurons of the neocortex[37]. We focused on genes expressed mainly in the cortical plate whose expression is altered in *Satb2fl/flNexCre* brains at E18 as compared to wild-type brains (Fig. S3). We hypothesized that the expression of a certain ligand-receptor pair would be changed in Satb2 mutants since our previous experiments suggested both cell-autonomous and non-cell-autonomous (Fig. S1) roles of Satb2 in neuronal migration and axon outgrowth. We reasoned that, the molecule acting as 'receptor' should be expressed in UL neurons where Satb2 has a cell-autonomous role while its potential putative 'ligand' could be expressed in both deep layers and upper layers.

We found one receptor-ligand pair that satisfied these criteria, Semaphorin7A/Integrinβ1 (Sema7A/Itgβ1). Expression patterns of both Sema7A and Itgβ1 were changed in the Satb2 mutant cortex. While Sema7A expression was reduced in UL neurons, Itgβ1 lost its lateral to medial gradient of expression within deep layers of the neocortex (Fig. 1A). We also re-analyzed published RNAseq data from a different Satb2 mutant[38] and found that expression of Sema7A remains reduced at P0 (Fig. 1B). In the embryonic cortex, the Sema7A Transcription Start Site (TSS) and intron 1 are well marked by histone modifications associated with transcriptional activity such as H3K27ac, H3K9ac, and H3K4me1/3 (Fig. 1C). ChIPseq peaks[39] for transcription factors NeuroD2, Tbr1 and Fezf2 have also been found near the TSS of Sema7A (Fig. 1C).

We performed Chromatin Immunoprecipitation using self-made Satb2 antibody[40] followed by quantitative real-time PCR (ChIP-qPCR) on E18 Cortical Lysates. We found Satb2 enriched in the 5' TSS region and not in the H3K4me1-marked intron 1 of Sema7A (Fig. 1C, D). This result was simultaneously confirmed in a Satb2-V5 ChIPseq[41] in the adult hippocampus, where a Satb2 peak was observed at the TSS of Sema7A (Fig. 1C, pink row).

To test whether Satb2 expression is sufficient to induce ectopic *Sema7A* transcription, we expressed a *Satb2 cDNA* construct in the cortical VZ/SVZ at E13 by IUE (Fig. 1E). This resulted in ectopic transcription of Sema7A mRNA at E16 in the VZ/SVZ as well as IZ regions of

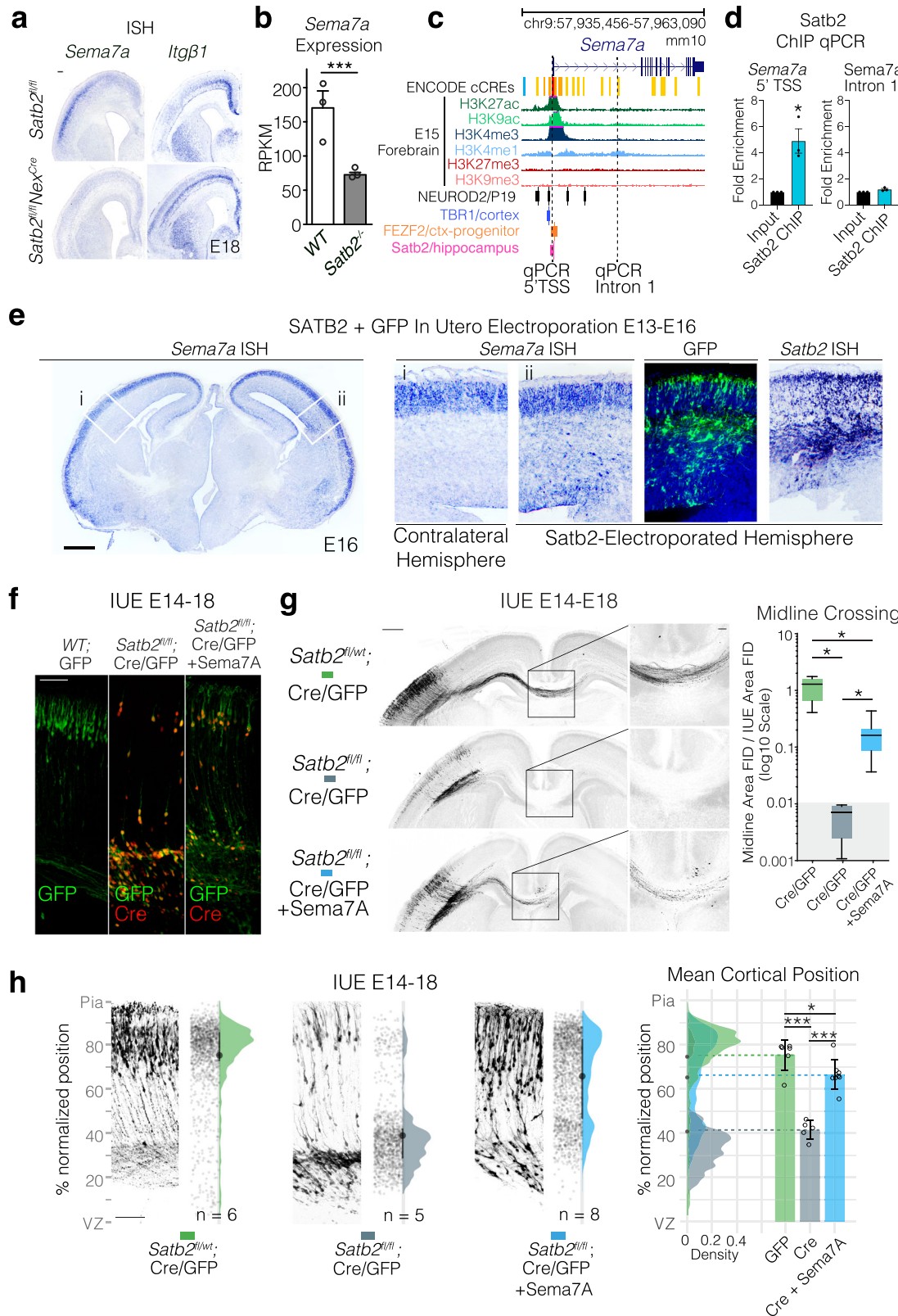

the neocortex, where it is not normally expressed (Fig. 1E). The less than 100% overlap of Satb2 overexpressing cells with Sema7A ectopic expression is likely because additional factors found only in post-mitotic neurons are required for efficient Sema7A expression, such as NeuroD2 (Fig. 1C). Indeed, we have recently come to understand that Satb2 is less a *de facto* transcription factor, and more akin to a chromatin looping factor[42].

We then asked whether restoration of *Sema7A* expression in *Satb2*-deficient neurons could restore defects of axonal specification and/or neuronal migration. Immunostaining showed that all GFP-expressing cells also expressed Cre but lacked Satb2 expression (Fig. 1F, S1D). Re-expression of *Sema7A* largely enabled midline crossing of Satb2-deficient axons in vivo (Fig. 1G). It also restored the laminar position of Satb2-deficient neurons (Fig. 1H). Our results

**Fig. 1 | Semaphorin7A is downstream of Satb2 and can restore migration and axon projection in Satb2 deficient neurons. a** In situ hybridization (ISH) of *Sema7A* and Itgβ1 expression in the Satb2 cortex, scale bar is 100µm. **b** Sema7A expression quantified as Reads Per Kilobase per Million mapped reads (RPKM), mean±SEM from $n_{cortices}$ WT = 3, Satb2KO = 3. DESeq2 q = 0.005283 (dataset from GSE68912). **c** UCSC genome browser view of Sema7A gene locus with annotation tracks: ENCODE cis-regulatory elements (cCREs), Histone ChIPseq from embryonic forebrain and ReMap ChIPseq tracks for NeuroD2, Tbr1 and Fezf2, and Satb2-V5 Hippocampal ChIPseq[41]. **d** Satb2 ChIP-qPCR fold enrichment (mean±SEM) of 5′TSS and Intron 1 Sema7A, $n_{cortices}$ WT = 3, *Satb2*[fl/fl]*Nex*[Cre] = 3. Unpaired t-test, 5′TSS; *p* = 0.0139, Intron 1; *p* = 0.656. **e** E16 wild-type brains electroporated at E13 with full-length *Satb2* full-length cDNA and GFP show ectopic expression of *Sema7A* RNA at the electroporation site. Scale bar 500 µm. **f** Overexpression of Sema7A into Satb2-deficient neurons restores migration cell autonomously. Immunostaining with GFP and Cre in *wild type* and *Satb2*[fl/fl] cortices after IUE with GFP (left), Cre/GFP (middle) and Cre/GFP +Sema7A (right). Scale bar is 100µm. **g** Re-expression of *Sema7A* in *Satb2* deficient cells partially rescues midline projections cell autonomously. *In utero* electroporations (IUE) into E14 embryos collected at E18 are shown with midline magnifications. In all conditions pNeuroD1-Cre + pCAG-FSF-GFP is abbreviated to Cre/GFP. Midline crossing was quantified by normalizing GFP Fluorescent Integrated Density (FID) at the midline to the FID of the electroporated area. Box and whisker plots represent whiskers as min-max with bounds of box at lower and upper quartiles and center at the median (right). $n_{brains}$ GFP = 5, Cre = 4, Cre + Sema7A = 8. Brown-Forsythe ANOVA with Dunnett's T3 multiple comparison test; GFP vs Cre $p_{adjusted}$ = 0.0197, GFP vs Cre + Sema7A $p_{adjusted}$ = 0.0357, Cre vs Cre +Sema7A $p_{adjusted}$ = 0.0161. Gray-ed out area at base of plot signifies measurements in this area are likely at the level of noise due to the complete absence of GFP fibers in Cre condition. Scale bar in the panoramic picture is 500µm, magnification is 100µm. **h** *Satb2* negative cells migrate into the cortical plate after *Sema7A* re-expression. Adjacent to example electroporation images, are raw data points corresponding to all cell positions, a half-violin showing the total cell distribution across $n_{brains}$ in that condition, where mean (point) and interquartile range (line) are plotted between raw data points and the half-violin. Cell distributions for the different conditions are shown overlaid on the right with mean and interquartile range per $n_{brains}$, along with the mean of means ±SD. One way ANOVA with Tukey's multiple comparisons. GFP vs Cre $p_{adjusted}$ < 0.0001, GFP vs Cre+Sema7A $p_{adjusted}$ = 0.0460, Cre vs Cre+Sema7A $p_{adjusted}$ < 0.0001. Scalebar at the bottom of GFP IUE = 100 µm. For simplicity, figures denote $p < 0.05$ as *; $p < 0.01$ as **; and $p < 0.001$ as ***. All source data are provided in the Source Data file.

show that Sema7A is a direct *Satb2* target that promotes radial neuronal migration and axon outgrowth during neocortical development.

## Sema7A is required for polarity acquisition and axon outgrowth

There are three crucial steps in the development of callosal projections: the specification of the axon and the leading process, axonal extension, and midline crossing. It is widely accepted that in the neocortex, the onset of axon extension coincides with the initiation of neuronal migration when a multipolar cell becomes polarized[4,43]. To better understand the phenotype of Satb2-deficient neurons (Fig. S2), we cultured dissociated Satb2-deficient (Satb2−/−) and wildtype (WT) neurons in vitro and characterized their morphology (Fig. 2). Future axons were identified as the longest neurite at DIV2 and by counterstaining for the axonal marker TAU at DIV4.

We observed that while Satb2−/− neurons do project axons in vitro, these axons are markedly shorter than WT cells at DIV2 and DIV4 (Fig. 2B, C). Furthermore, Satb2−/− cells possess more primary neurites than WT cells at DIV2, which appears to result in increased branching (End neurites) by DIV4 (Fig. 2A, C). While Satb2−/− neurons do project axons in vitro, these axons are markedly shorter than WT and these neurons do not achieve complete polarization (Fig. 2B, C). While the length of the longest neurite is unaffected at DIV2, Sema7A re-expression restored the number of primary and end neurites to WT levels (Fig. 2A, C). By DIV4, most Sema7A-rescued Satb2−/− neurons also showed recovered TAU + /MAP2- axon lengths.

Previous studies have reported that a young neuron becomes polarized when the microtubule organizing center, the centrosome, positions itself together with the Golgi apparatus in front of the neurite that will become the axon, and prior to migration moves to specify the leading process[44–46]. To examine the possible role of Satb2 in mediating polarization, we measured the positions of the centrosome and Golgi organelles with respect to the closest process. At DIV2 in wild-type neurons, the centrosome lies at the base of and in line with the longest neurite along the 0-180° 'polarity' axis (Fig. 2D, left). Loss of Satb2 was associated with a disturbance in the position of the centrosome, with this structure being mostly found between neurites, resulting in angles that are further away the polarity axis (Fig. 2D, middle).

We then analyzed the role of Sema7A in regulating neuronal morphology and polarity downstream of Satb2 by overexpression of Sema7A in these cells. We found that the re-expression of *Sema7A* in *Satb2*−/− neurons shifted the centrosome angle back towards the polarity axis (Fig. 2D, right). Collectively, our results show that Sema7A controls neuronal polarity and outgrowth as well as the initiation of migration downstream *of Satb2*.

## Sema7A membrane localization and dimerization are required to mediate cell-autonomous effects

Since Sema7A lacks an intracellular domain and is reported to act as a ligand in trans[47], it was difficult to envisage how it mediated a cell-autonomous rescue in Satb2 deficient neurons. One possibility was that Sema7A was being secreted to act in an autocrine fashion on the cell expressing it. We therefore repeated genetic rescue experiments with a mutant form of Sema7A, where we removed the GPI membrane anchor (ΔMEM). Expression of the secreted version of Sema7A (ΔMEM) was neither capable of restoring migration, nor axon elongation in Satb2 mutants (Fig. 3B, C). This indicates that Sema7A requires membrane attachment to function cell autonomously downstream of Satb2. Another possibility is that Sema7A acts by complexing in *cis* with other transmembrane receptors capable of reverse signaling to effect cytoskeletal changes cell autonomously. The seven-bladed beta-propeller SEMA domain has been shown to mediate dimerization of Semaphorins, a characteristic necessary for their function[48,49], as well as interaction with other receptors (eg Nrp1[50]). Deletion of this important domain also interferes with Sema7A function downstream of Satb2 (Fig. 3B, C) indicating that dimerization is essential for Sema7A function. We also generated a version of Sema7A where we mutated the conserved Arginine Glycine Aspartate (RGD) motif required for Integrin binding[51], (KCE mutant, Fig. 3A). Mutation of this site interestingly disrupted the ability of Sema7A to promote axon outgrowth, and to a lesser extent migration, downstream of Satb2. We obtained a similar result upon deletion of the structural IG or the PSI domain that in other SEMA members is required for the correct positioning of the ligand-binding site[52], suggesting differences in the signaling roles of Sema7A during these two processes. Together these data show that Sema7A is acting as a membrane-associated receptor cell autonomously, and given its lack of an intracellular domain it was most likely in complex with other transmembrane receptors.

## Sema7A acts as a co-receptor of Sema4D

We asked whether Sema7A could heterodimerize with other semaphorin family members that would then be able to transmit signals into the expressing cell in a cell-autonomous manner. Members of Semaphorin Class 4 and Class 6 had recently been shown to be involved in reverse signaling during cell polarization and migration[28–31]. In addition, the secreted Drosophila Sema-2a and 2b use the transmembrane Sema-1a to initiate signaling[29,53,54]. We focused on semaphorins that

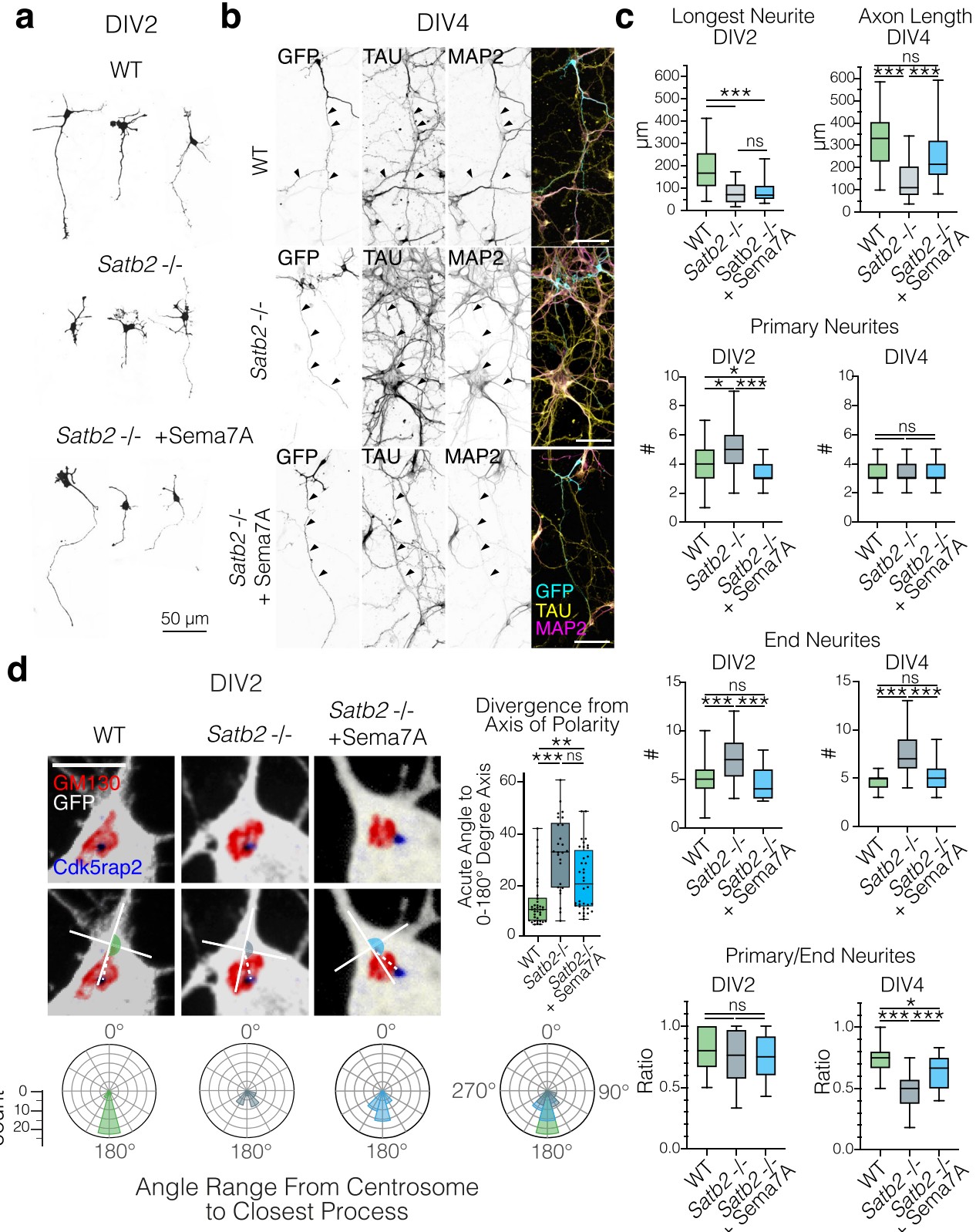

contain a cytoplasmic domain and could thereby transmit signals into the expressing cell, in a cell-autonomous fashion. We also reasoned that the expression should overlap with that of Sema7A and should remain in Satb2 deficient cortex (Fig. S3). We first performed co-immunoprecipitation experiments (Co-IP) in HEK293T cells using a C-terminal GFP-tagged Sema7A and N-terminal Myc-tagged versions of Sema5A, Sema6A, and Sema4D – three putative receptors that meet

above-mentioned criteria (Fig. 4A). We detected strong binding affinity between Sema7A and Sema4D and a weak interaction with Sema6A (Fig. 4A). Notably, Sema4D is also expressed in the developing cortical plate alongside Sema7A (Fig. 4B). While Sema7A is reduced in the Satb2 mutant, Sema4D is modestly upregulated (Fig. S3).

Sema4D is known to form homodimers[55]. To test whether Sema7A competes with Sema4D for dimerization, we additionally cloned a

**Fig. 2 | Sema7A restores the polarization and axon outgrowth in *Satb2*-deficient neurons also in vitro. a** Examples of WT, Satb2 deficient, and Sema7A rescue conditions in E14 primary cortical neurons after 2 days in vitro (DIV2). **b** After 4 days in vitro (DIV4), axons can be identified in all conditions by their enrichment of TAU-1 and depletion of MAP2. Scale bar = 50 µm. **c** Box & whisker plots (whiskers: min, max, box spanning the interquartile range, and center line at median) of neuronal morphology at DIV2 (left column) and DIV 4 (right column). Data were non-normally distributed (D'Agostino & Pearson, Shapiro-Wilk tests). For all tests, Kruskal–Wallis test followed by Dunn's Multiple comparison was used, where adjusted $p$ values < 0.001 = ***, <0.01 = **, <0.05 = *. DIV2 longest neurite $n_{neurons}$ WT = 48, Satb2−/− = 54, Satb2−/− + Sema7A n = 37; DIV2 $n_{neurons}$ primary/end neurites WT = 40, Satb2−/− = 39, Satb2−/− + Sema7A = 37. DIV4 axon length $n_{neurons}$ WT = 20, Satb2 −/− = 39, Satb2−/− + Sema7A = 37. DIV4 $n_{neurons}$ primary/end neurites WT = 21, Satb2−/− = 39, Satb2−/− + Sema7A = 37. A full list of $p$ values can

be found in Supplementary Data S2. **d** Loss of Satb2 is associated with a random distribution of the centrosome which can be restored by Sema7A. (Top row): Representative images of DIV2 primary cortical neurons stained with the golgi marker Gm130 and centrosomal marker Cdk5rap2 (scale bar 25µm). Radar plots depict the circularized histograms of angle counts, where 360 degrees are binned in 30-degree increments. Measured blind, the polarity angle was measured using a standardized 100 × 100 pixel cross placed with the parallel axis centered in the process and the perpendicular axis touching the Golgi. Data were non-normally distributed (D'Agostino & Pearson, Shapiro–Wilk tests). Kruskal Wallis test followed by Dunn's Multiple comparison $p_{adjusted}$ GFP vs Cre <0.0001, $p_{adjusted}$ GFP vs Cre + Sema7A = 0.0026, $p_{adjusted}$ Cre vs Cre + Sema7A = 0.0794. $n_{neurons}$ WT = 35, Satb2−/− = 25, Satb2−/− + Sema7A = 35. For simplicity, panels denote $p < 0.05$ as *; $p < 0.01$ as **; and $p < 0.001$ as ***. All source data are provided in the Source Data file.

version of Sema7A that contains an HA tag at an exposed loop of the Sema domain at codon 352, (HA-Sema7A) and a Sema4D with a C-terminal (cytoplasmic) flag tag. Increasing amounts of Sema7A resulted in decreased amounts of Sema4D-Flag co-immunoprecipitated with N-Myc-Sema4D, indicating that Sema7A competes with Sema4D monomers for dimerization (Fig. 4C). We observed 150 kDa and 120 kDa forms of Sema4D similar to previous reports[56]. 120 kDa and 150 kDa forms of Sema4D are both 'full length' given the detection of the C terminal tag, however, in the presence of N-Myc-Sema4D, Sema4D-flag is observed more prominently as the 120 kDa form, which is also the preferred form pulled down by co-immunoprecipitation in flag lysis buffer (Fig. 4C). This suggests the existence of differential post-translational modification of the homodimerized form of Sema4D as compared to Sema4D present in Sema4D:Sema7A heterodimers.

We used the published crystal structures of Sema4D[57] and Sema7A[47] to model the Sema7A:Sema4D heterodimer. We found that superimposition of one Sema4D molecule over one Sema7A molecule from the published Sema7A homodimer, resulted in only a minor deviation (RMSD:0.64 Å) in the Cα atoms of the proteins (Fig. 4D, further supporting the existence of a physiological competitive interaction between these two proteins.

Given N-Myc-Sema4D (addgene #51599) uses a chimeric signal peptide and produces very little of the 150 kDa form of Sema4D[58], we cloned Sema4D to contain a myc tag after its endogenous signal peptide based on structural data (termed sp-Myc-Sema4D). This construct successfully expresses both ~120 kDa and 150 kDa forms of Sema4D (Fig. 4E).

Next, we asked if Sema7A:Sema4D complexes are formed in the neocortex during development. Immunoprecipitation of endogenous Sema4D from E18.5 cortical lysates resulted in co-immunoprecipitation of endogenous Sema7A, demonstrating that Sema7A:Sema4D complexes form in vivo (Fig. 4F). Furthermore, we visualized the subcellular localization of this interaction in vivo, using Proximity Ligation Assay (PLA) in wild type cortical neurons transfected with sp-Myc-Sema4D and HA-Sema7A (Fig. 4G–L). Indeed, we observe the direct interaction of HA-Sema7A and sp-Myc-Sema4D in neurons. At DIV2, the Sema4D-Sema7A heterodimer can be seen in the soma and axon hillock (Fig. 4G, H). In more mature neurons at DIV4, PLA signal is further enriched in the growing axon and is observed at the tips of filopodia, branch points and with an enrichment at axonal growth cones (Fig. 4I–L).

## Sema4D is required for cell-autonomous Sema7A-mediated neuronal migration and axonal growth

To address whether Sema7A and Sema4D function together in promoting neuronal migration and axonal growth cell autonomously, we conducted loss-of-function experiments in vivo. We first addressed whether the role of Sema7A downstream of Satb2 in promoting migration and axon outgrowth was dependent on the function of

Sema4D. This was carried out by downregulating Sema4D expression using a specific shRNA, while restoring Sema7A expression in Satb2-deficient neurons in vivo.

As previously observed, re-expression of Sema7A improved the laminar position (Fig. 5A) and callosal projections of Satb2-deficient neurons (blue condition, Fig. 5B, C). Downregulation of Sema4D, however, prevented Sema7A-mediated rescue of both migration (Fig. 5A) and axonal outgrowth in the *Satb2*-deficient neurons (Fig. 5B, C, red condition). Additional controls for this experiment can be seen in Fig. S4.

We also tested whether downregulation of Sema4D or Sema7A by shRNA would phenocopy cell autonomous deletion of Satb2. In a similar design, we introduced a construct expressing *shRNA* against Sema7A into the E14 cortex and analyzed the brains at E18. Indeed, Sema7A knockdown resulted in reduced projection of axons into the contralateral hemisphere (Fig. 5D, E) and a migration deficit of upper layer neurons (Fig. 5F). Similar experiments using shRNA against Sema4D, produced even stronger phenotypes, with strong disruption of both migration and axon growth (Fig. 5D–F).

Our hypothesis of Sema7A was based on its interaction in cis with a transmembrane receptor that could thereby mediate reverse signaling. We therefore addressed if the intracellular domain (ICD) of Sema4D is required for it function in neuronal migration and axonal growth. To test this, we downregulated Sema4D using a specific shRNA, and at the same time overexpressed either full-length Sema4D, or Sema4D lacking the ICD (Sema4D-ΔIC). Only the construct encoding the full cDNA could rescue migration and moderately rescue axon outgrowth caused by Sema4D loss-of-function, while Sema4D-ΔIC was unable to repair these defects (Fig. 5D–F, light and dark purple conditions). Together, these data show that the intracellular domain of Sema4D is required for the cell-autonomous effects of Sema7A on the radial migration and axonal outgrowth of upper-layer neurons in the developing neocortex.

## Human SEMA4D-497P mutation inhibits trafficking to the membrane and growth cone

Over the course of this study, we identified a patient presenting with generalized tonic-clonic seizures that has a de novo mutation in SEMA4D (see case report in supplemental note 1). Exome sequencing revealed a non-synonymous mutation of adenosine to cytosine resulting in a glutamine (Q) to proline (P) substitution at codon 497 (Fig. 6A). Given that the Q497 residue is involved in the formation of a beta sheet on a propeller of the sema domain (Fig. 6B), the sudden inclusion of a proline ring was expected to alter secondary structure. To predict how this mutation may affect protein folding, we used the most recent release of alphafold2[59] to compare predicted structure with the known crystal structure of SEMA4D (PDB: 1OLZ). Interestingly, the predicted structure of wildtype SEMA4D (Fig. 7B, middle panel) and SEMA4D-497P are very close to the 1OLZ crystal structure (Fig. 6B, upper panel). In both WT and 497P predictions,

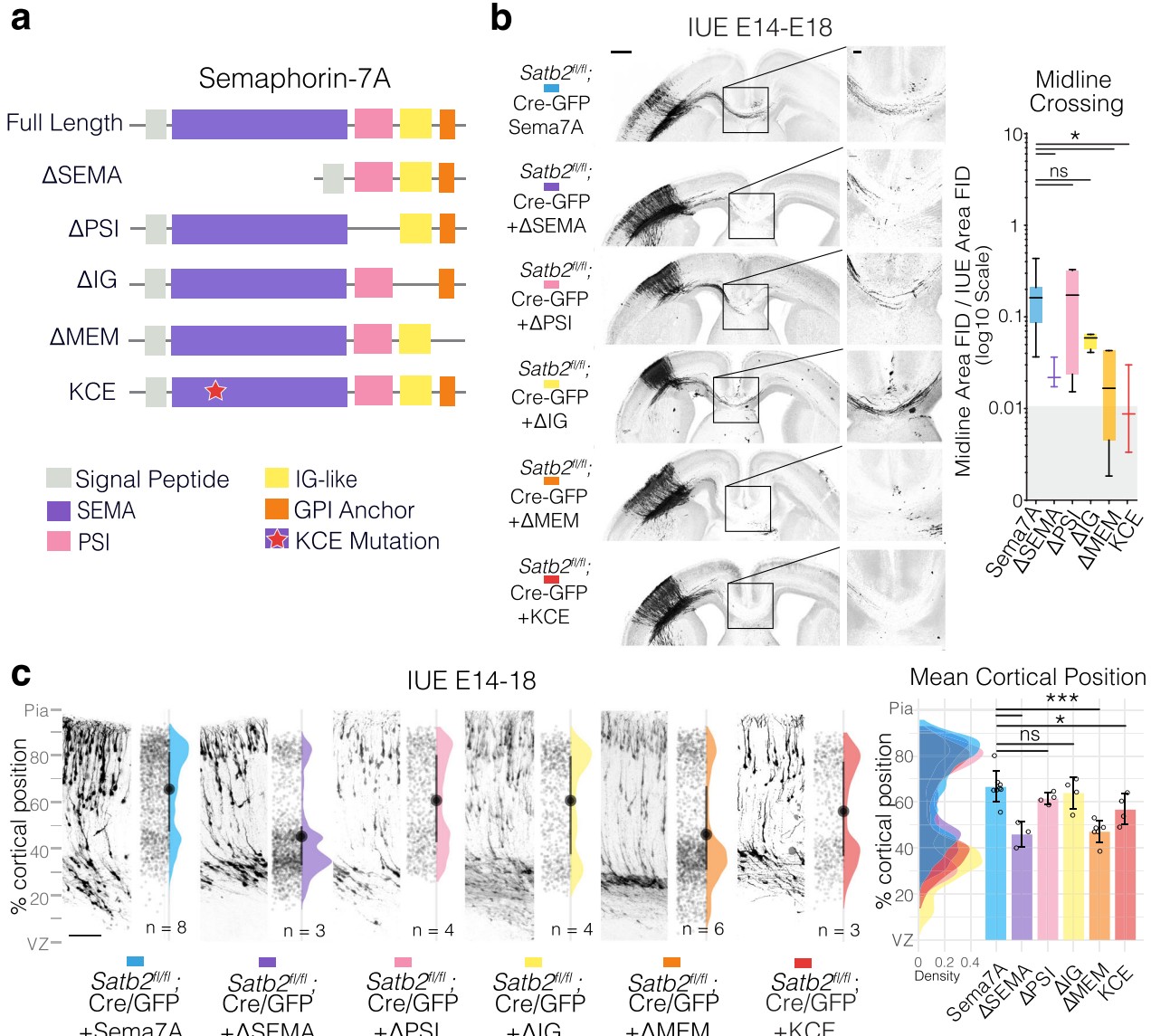

**Fig. 3 | Semaphorin 7A Domains associated with membrane localization and dimerization are required for migration and axon outgrowth. a** Schematic of Sema7A domains and deletion mutants. The scheme shows all currently annotated domains of Sema7A and depicts the deletion constructs generated. Sema domain is depicted in purple, the Plexin-Semaphorin-Integrin (PSI) domain in pink, the immunoglobulin domain (IG) in yellow and the Glycosyl-Phosphatidyl-Inositol (GPI) membrane anchor is depicted in orange. The Arginine-Glycine-Aspartic acid (RGD) reported to be important for Integrin binding was mutated to Lysine-Cysteine-Glutamic Acid (KCE) and is depicted as a red star. **b** Panoramas of *Satb2^{fl/fl}* IUE brains with the indicated Sema7A deletion constructs together with pCAG-FSF-GFP and pNeuroD1-Cre and midline magnifications and quantifications. Full Length Sema7A panorama is reproduced here from Fig. 1 for comparison. Midline crossing is quantified as described in Fig. 1. Box and whisker plot whiskers as min-max with box bounds at lower and upper quartiles and center line on the median. Grey-ed out area at base of plot signifies measurements in this area are likely at the level of noise due to the complete absence of GFP fibers in Cre only condition (Fig. 1). Brown-Forsythe ANOVA with Dunnett's T3 multiple comparisons test. $p_{adjusted}$ Sema7A vs ΔSEMA = 0.0457, $p_{adjusted}$ Sema7A vs ΔMEM = 0.0416, $p_{adjusted}$ Sema7A vs KCE = 0.0342. $n_{brains}$ Cre + Sema7A (from Fig. 1) = 8, $n_{brains}$ Cre + Sema7A-ΔSEMA = 3, $n_{brains}$ Cre + Sema7A-ΔPSI = 4, $n_{brains}$ Cre + Sema7A-ΔIG = 4, $n_{brains}$ Cre + Sema7A-ΔMEM = 6, $n_{brains}$ Cre + Sema7A-KCE = 3. Scale bar in panoramic picture is 500 μm, while scale bar in midline magnification is 100 μm. **c** Sema7A deletion mutant migration profiles from E14-18 IUEs, presented in the same format described in Fig. 1. Cre + Full length Sema7A is reproduced here from Fig. 1 for comparison. Scalebar bottom left of Cre+Sema7A IUE is 100μm. (Right): Cell distributions for the different conditions are shown overlaid on the right with mean cortical position per $n_{brains}$, along with the mean of means ±SD. One way ANOVA with Dunnett's multiple comparisons test. $p_{adjusted}$ Sema7A vs ΔSEMA = 0.0001, $p_{adjusted}$ Sema7A vs ΔMEM < 0.0001, $p_{adjusted}$ Sema7A vs KCE = 0.0493. $n_{brains}$ Cre + Sema7A (from Fig. 1) = 8, $n_{brains}$ Cre + Sema7A-ΔSEMA = 3, $n_{brains}$ Cre + Sema7A-ΔPSI = 4, $n_{brains}$ Cre + Sema7A-ΔIG = 4, $n_{brains}$ Cre + Sema7A-ΔMEM = 6, $n_{brains}$ Cre + Sema7A-KCE = 4. For simplicity, panels denote $p < 0.05$ as *; $p < 0.01$ as **; and $p < 0.001$ as ***. All source data are provided in the Source Data file.

upper segments of the sema domain form low confidence loops, due to a limitation of alphafold2 (Fig. S5). However, the beta-sheet containing codon 497 is consistent between 1OLZ and the prediction for SEMA4D, so the comparison between 497 mutation structure to wildtype is useful at this level. While the beta-sheet containing P497 is predicted to form correctly, the beta-sheet below this is absent in the predicted model of SEMA4D-Q497P. This lower beta sheet normally resides in close proximity to glycosylation site N77 (Fig. 6B, lower panel zoom). It is also possible that steric hindrance from the Q497P mutation could affect normal glycosylation at N49 and N419 or ubiquitination at K505 or K81 (Fig. 6B, lower panel zoom).

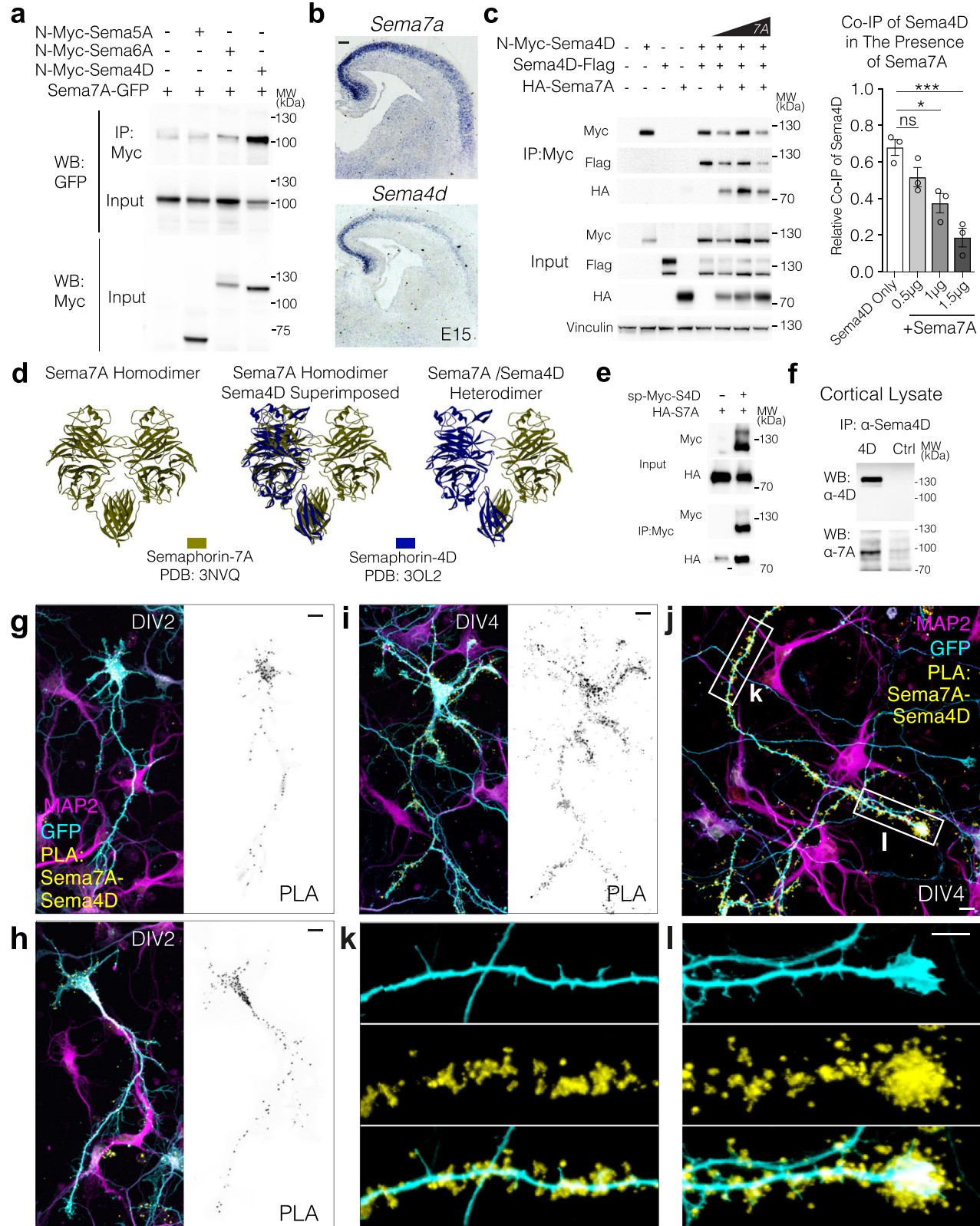

We asked if SEMA4D-497P can still form homo- and heterodimers after immunoprecipitation of proteins overexpressed in HEK293T cells. The Q497P mutation did not interfere with SEMA4D homodimerization or heterodimerization with mouse Sema7A (Fig. 6C). However, we observed a shift of the Sema4D bands in an SDS-PAGE gel. While wildtype SEMA4D normally migrates in two sizes (~150 kDa and

~120 kDa), the heavier form of the protein (~150kDA) was largely absent in SEMA4D-497P (Fig. 6C).

In line with our structural predictions, we hypothesized that the two forms of Sema4D could correspond to different maturation states or glycosylation states of the protein. To test this hypothesis, we incubated SEMA4D overexpressing lysates with de-glycosylation

**Fig. 4 | Sema4D binds with high affinity to Sema7A. a** Sema7A binds to Sema4D and Sema6A. HEK293T cells transfected with the GFP-tagged Sema7A (this paper) and the indicated Myc-tagged Sema family members: *N*-Myc-Sema4D (addgene # 51599, -120 kDa), N-myc-Sema5A (-65 kDa), and N-myc-Sema6A (-120/130 kDa). **b** In Situ *Hybridization* (ISH) at E15 in wild-type mouse cortex shows the expression of both Semaphorins in the cortical plate. Scale bar= 100μm. **c** Co-IP of Sema4D-flag from myc-Sema4D:Sema4D-flag homodimers in the presence of increasing concentrations of HA-tagged Sema7A in HEK cells. One way ANOVA with Dunnett's multiple comparison test. $p_{adjusted}$ Sema4D only vs 1.0 μg Sema7A = 0.0068, $p_{adjusted}$ Sema4D only vs 1.5 μg Sema7A = 0.0003. $n_{bio.replicates}$ = 3. **d** Computational modeling of structural alignment of Sema4D:Sema7A heterodimer using published crystal structures. **e** Co-immunoprecipitation of HA-Sema7A using sp-myc-Sema4D in HEK cells. **f** Endogenous Co-immunoprecipitation of Sema7A using anti-Sema4D antibody from mouse cortex. **g–l** Localization and distribution of Sema4D-Sema7A complexes in primary cortical neurons. Primary E14 neurons transfected with sp-Myc-Sema4D, HA-Sema7A, and GFP were fixed at DIV2 and DIV4 and subcellular localization of semaphorin complexes determined by Proximity Ligation Assay (PLA). GFP signal (cell fill) can be seen in cyan, Sema4D-Sema7A PLA signal in yellow, and MAP2 in magenta. Scale bars in (**g–l**) are 10μm. For simplicity, panels denote $p < 0.05$ as *; $p < 0.01$ as **; and $p < 0.001$ as ***. All source data are provided in the Source Data file.

enzymes. De-glycosylation with O-Glycosidase resulted in the loss of the 150 kDa migrating form of SEMA4D and the presence of only a 120 kDa form, indicating that this higher migrating form results from *O*-linked glycosylation of the protein. Treatment of SEMA4D with PNGase F, which cleaves almost all glycosylation types, results in further downwards shift of the SEMA4D band to 110 kDa, indicating additional N-glycosylation is present on SEMA4D. Endo H, which cannot cleave complex glycans, results in a mixed picture of 150, 120 and 110 kDa migrating forms of SEMA4D.

We then analyzed glycosylation on the mutated form of SEMA4D. SEMA4D-Q497P runs at 120 kDa and lacks the 150 kDa migrating form that was associated with O-linked glycosylation in wildtype SEMA4D. Indeed SEMA4D-Q497P was insensitive to O-Glycosidase indicating that unlike the wildtype form, SEMA4D-Q497P lacks O-linked glycosylation. SEMA4D-Q497P is still sensitive to PNGase F, showing that the protein is nevertheless modified by other glycosylation types. The incomplete glycosylation of SEMA4D-Q497P is also evident upon incubation with Endo H, which only yields a 110 kDa form, compared to the intermediate 120 kDa fragments observed when digesting WT SEMA4D (Fig. 6D, WT blue arrow vs Q497P empty blue arrow). Notably however, treatment with PNGase F, which should cleave nearly all glycosylation, still results in two prominent WT bands, reduced to -120 kDa and -110 kDa, suggesting that additional post-translational modifications contribute to the difference in protein size.

Given that presence of the SEMA4D receptor at the plasma membrane is required for its biological function, and glycosylation is known to be important in protein localization[60], we used surface biotinylation followed by avidin pull down to address the role of Sema7A and of the Q497P mutation in regulating the subcellular localization of SEMA4D (Fig. 7A). Interestingly, only the 150KDa form of SEMA4D became biotinylated, indicating that this O-linked glycosylated form is the membrane-localized form of the protein. We observed that co-expression of Sema7A increased the proportion of surface-localized SEMA4D three-fold, while mutation of SEMA4D at residue 497 abolished its localization to the cell surface, indicating that both interaction with Sema7A and correct glycosylation are necessary for correct plasma membrane localization of SEMA4D.

To further characterize this effect, we nucleofected primary neurons with mouse HA-Sema7A and human sp-myc-SEMA4D or sp-myc-SEMA4D-Q497P and performed PLA to observe the location of the semaphorin complex in situ (Fig. 7B). We observed that while SEMA4D:Sema7A complexes are normally found on the outer side of the cell membrane and the growth cone (as per Fig. 4), SEMA4D-497P:Sema7A complexes are predominantly found intracellularly in the soma and are absent from the growth cone (Fig. 7Bi-iv), indicating that mutation of residue 497 causes loss of SEMA4D:Sema7A complexes from the plasma membrane.

Finally, we analyzed the functional consequences of mutation of Sema4D during neocortical development. Overexpression of SEMA4D-Q497P in the developing neocortex by *in utero* electroporation, disrupted radial migration and neurons collected in the lower portions of the neocortex (Fig. 7C, D). Furthermore, overexpression of SEMA4D-Q497P prevented axon projection across the corpus callosum (Fig. 7C).

Thus, mutation of Sema4D at Q497 phenocopies downregulation of either Sema7A or Sema4D. Given that the mutation does not interfere with Sema4D homo- or heterodimerization (with Sema7A), together this data suggests that the dominant negative actions of this mutation result from disruption of the plasma membrane localization of the Sema4D:Sema7A complex.

## Discussion

In this study we observed that the transcription factor Satb2 promotes neuronal migration and axon outgrowth in the developing neocortex by directly inducing expression of the semaphorin family member Sema7A (Figs. 1, 7E). Sema7A is a GPI-linked protein, which initiates reverse signaling to exert its biological functions downstream of Satb2 by heterodimerizing with Sema4D, a transmembrane Semaphorin family member. Sema7A:Sema4D complexes can be found at neurites and are enriched in the growth cone (Fig. 4). Membrane localization of the Sema4D is dependent on heterodimerization with Sema7A and on correct O-linked glycosylation. The importance of promoting the membrane localization of Sema4D is emphasized by the identification of a human mutation in Sema4D, Q497P associated with tonic-clonic seizures. Q497P interferes with the membrane localization of the Sema7A:Sema4D complex most likely by disrupting O-linked glycosylation events on Sema4D. Q497P Sema4D thereby acts in a dominant negative fashion to interfere with neuronal migration and axon projection during neocortical development (Fig. 7).

Thus, we have identified a key role for the Satb2-target and GPI-linked semaphorin, Sema7A, in initiating reverse signaling that promotes correct cellular polarity and drives cell-autonomous migration and axon elongation of cortical projection neurons (Fig. 7E).

Sema7A lacks a cytoplasmic domain and exerts its biological function in the neocortex by heterodimerizing in cis with the transmembrane Semaphorin, Sema4D. Previous studies have identified Sema7A to be important for olfactory bulb, olfactory- and thalamo-cortical tracts, as well as for hippocampal and cortical axonal outgrowth and branching[25,51,61]. However, these studies assumed that Sema7A acts only as a classical ligand, initiating signaling in a neighboring cell after binding in trans to one of its receptors, Integrinβ1 or PlexinC1. The interpretation of these previous studies may need to be adjusted to account for the existence of Semaphorin:Semaphorin interactions and their ability to engage in reverse signaling as described here. In addition to the strong interaction of Sema7A with Sema4D, we also detected weaker binding between Sema7A and Sema6A, suggesting that other Semaphorin signaling combinations and reverse signaling cascades may exist. Until now, reverse signaling has only been demonstrated for a handful of transmembrane semaphorins acting alone through their own cytoplasmic domains; Sema1a[29], Sema4A[31], Sema4C[62], Sema6A[28], Sema6D[30]. Further experiments are needed to characterize the full extent of Semaphorin:Semaphorin interactions and their capacity to carry out reverse signaling.

Our study provides evidence that Sema4D functions downstream of Sema7A to promote radial migration and axon outgrowth of upper layer neurons in the neocortex. Downregulation of either Sema7A or

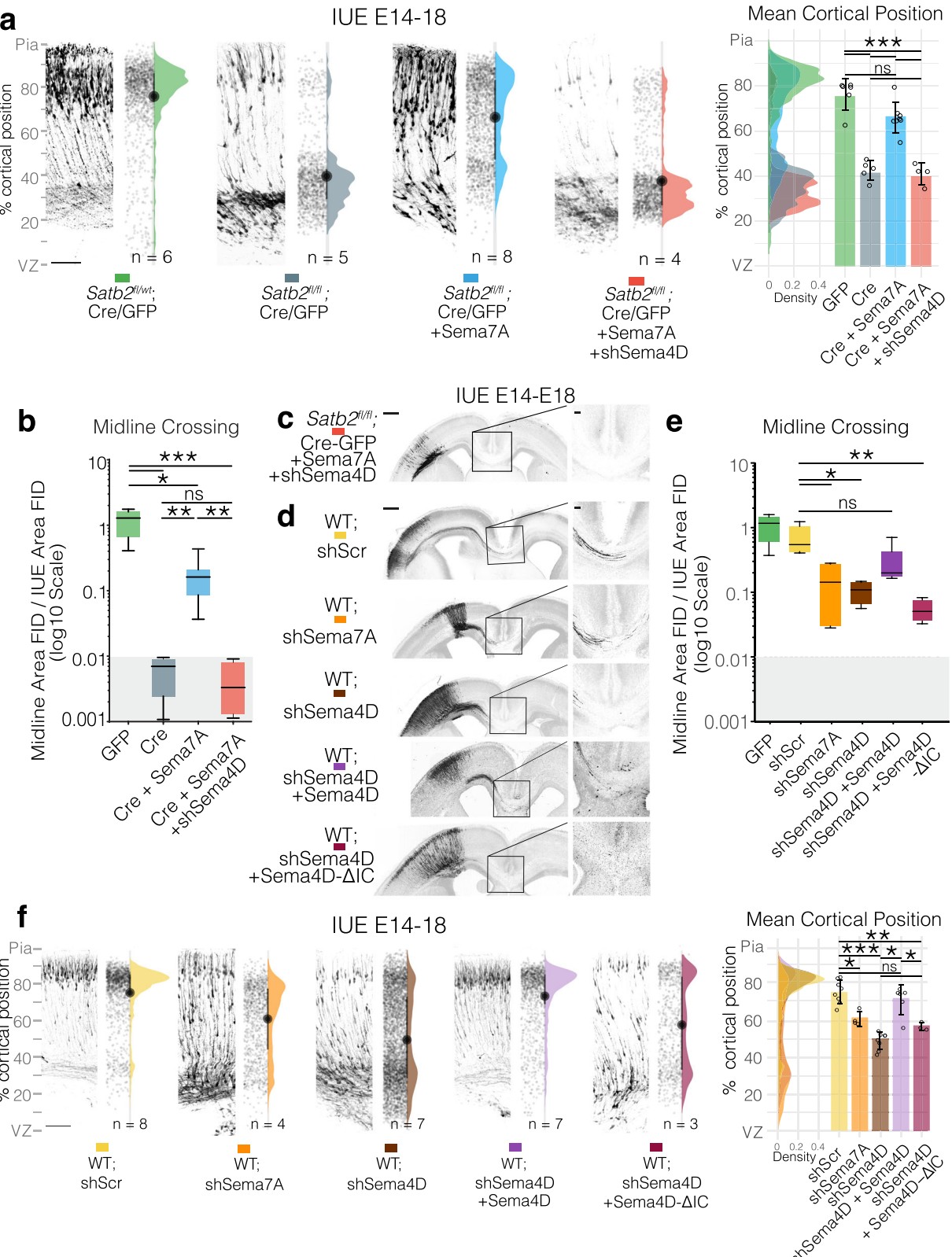

Sema4D is sufficient to disrupt radial neuronal migration and axon outgrowth along the corpus callosum (Fig. 5). Moreover, while axonal and migratory defects induced by loss of the transcription factor Satb2 are restored by overexpression of Sema7A, this rescue is still dependent on expression of Sema4D. (Figs. 1 and 5).

Sema4D and Sema7A directly interact to form a complex during neocortical development (Fig. 4). Our computational protein structure model suggests that Sema4D and Sema7A interact via the SEMA domain. This is concordant with previous studies that address the structure and function of this domain[16,27,63]. The SEMA domain also mediates interaction of Semaphorins in trans to Plexins and neuropilins[64]. Heterotetramer complexes consisting of Semaphorin homodimers and dissociated Plexin monomers have been described[57,65]. Similarly, Sema7A:Sema4D heterodimers may also be

**Fig. 5 | The Cytoplasmic Domain of Sema4D is required for cell-autonomous migration and axon outgrowth. a** Simultaneous downregulation of Semaphorin 4D by shRNA reverses Semaphorin 7A rescue of neuronal migration in Satb2-deficient neurons. Neuronal migration profiles from E14-18 IUEs, presented in the same format described in Fig. 1. GFP, Cre and Cre + Sema7A are reproduced from Fig. 1 here for comparison. Scalebar in the bottom of GFP IUE is 100μm. Cell distributions for the different conditions are shown overlaid on the right with mean cortical position per n$_{brains}$, along with the mean of means ±SD. One way ANOVA with Tukey's multiple comparisons test. GFP vs Cre $p_{adjusted}$ < 0.0001, GFP vs Cre +Sema7A $p_{adjusted}$ = 0.0604, GFP vs Cre+Sema7A+shSema4D $p_{adjusted}$ < 0.0001, Cre vs Cre+Sema7A $p_{adjusted}$ < 0.0001, Cre vs Cre+Sema7A+shSema4D $p_{adjusted}$ = 0.9816, Cre+Sema7A vs Cre+Sema7A+shSema4D $p_{adjusted}$ < 0.0001. **b** Quantification of midline crossing after simultaneous downregulation of Semaphorin 4D in Sema7A-rescued Satb2-deficient neurons. GFP, Cre and Cre+Sema7A images and data are reproduced from Fig. 1. Brown-Forsythe ANOVA with Dunnett's T3 multiple comparisons test. GFP vs Cre $p_{adjusted}$ = 0.0328, GFP vs Cre+Sema7A $p_{adjusted}$ = 0.0589, GFP vs Cre+Sema7A+shSema4D $p_{adjusted}$ = 0.0326, Cre vs Cre +Sema7A $p_{adjusted}$ = 0.0292, Cre vs Cre+Sema7A+shSema4D $p_{adjusted}$ = 0.9592, Cre +Sema7A vs Cre+Sema7A+shSema4D $p_{adjusted}$ = 0.0277. Migration n$_{brains}$ GFP = 6, Cre = 5, Cre + Sema7A = 8, Cre + Sema7A+ shSema4D = 4. **c** shows Cre+Sema7A +shSema4D midline panorama example. Panorama scalebar = 500μm, magnification scalebar = 100μm. **d** Midline panoramas of Sema4D and Sema7A shRNA electroporations and magnifications. Control scrambled shRNA (shScr), shRNA against Sema4D, shRNA against Sema4D + Sema4D cDNA (Sema4D Overexpression, OE), or shRNA against Sema4D plus a version of Sema4D cDNA lacking the intracellular domain -IC (Sema4D-ΔIC) were introduced into the cerebral cortex at E14 and axons crossing the midline were observed at E18. Scale bar in panoramas is 500 μm, while scale bar in midline magnification is 100 μm. **e** Quantification of midline crossing of shRNA electroporations. Kruskal-Wallis with Dunn's multiple comparison test. scrambled shRNA (shScr) vs shSema7A $p_{adjusted}$ = 0.0404, shScr vs shSema4D $p_{adjusted}$ = 0.0253, shScr vs shSema4D+Sema4D OE $p_{adjusted}$ = 0.6127, shScr vs shSema4D + Sema4D-ΔIC $p_{adjusted}$ = 0.0017. In box and whisker plots (**b, e**) whiskers represent min-max with box bounds at lower and upper quartiles and center line at the median. Midline n$_{brains}$ GFP = 5, Cre = 4, Cre + Sema7A = 8, Cre + Sema7A + shSema4D = 4, shScr = 5, shSema7A = 4, shSema4D = 4, shSema4D + Sema4D OE = 7, shSema4D + Sema4D-ΔIC = 4. **f** Neuronal migration profiles of after downregulation of Sema4D from E14-18 IUEs, presented in the same format described in Fig. 1. Scalebar in the bottom of shScr IUE is 100μm. Representative electroporations are shown adjacent to raw data points and half violin plots of the total distribution with mean of means±SD cortical position on the right. One way ANOVA with Tukey's multiple comparisons. scScr vs shSema7A $p_{adjusted}$ = 0.0069, scScr vs shSema4D $p_{adjusted}$ < 0.0001, shScr vs shSema4D+Sema4D OE $p_{adjusted}$ = 0.8358, shScr vs shSema4D+Sema4D-ΔIC $p_{adjusted}$ = 0.001, shSema4D vs shSema4D + Sema4D OE $p_{adjusted}$ < 0.0001, shSema4D vs shSema4D+Sema4D-ΔIC $p_{adjusted}$ = 0.3605, shSema4D-Sema4D OE vs shSema4D+Sema4D-ΔIC $p_{adjusted}$ = 0.0115. Migration n$_{brains}$ shScr = 8, shSema4D = 8, shSema7A = 4, shSema4D + Sema4D OE = 6, shSema4D + Sema4D-ΔIC = 3. For simplicity, panels denote $p$ < 0.05 as *; $p$ < 0.01 as **; and $p$ < 0.001 as ***. All source data are provided in the Source Data file.

involved in forward signaling through interaction with other receptors on neighboring cells.

Our data indicate that heterodimerization of Sema4D with Sema7A is necessary for the plasma membrane localization of Sema4D and downstream biological function (Fig. 7). Plasma membrane localization of Sema4D may bring it in proximity to downstream effectors. Few direct binding partners for the Sema4D intracellular domain have been described. In T lymphocytes, an unknown Ser/Thr kinase has been reported to interact with the Sema4D intracellular domain[56,66]. In addition, direct Sema4D ligation experiments in non-neuronal cell types suggest that ERK MAP kinase, cofilin and Rac GTPase signaling, pathways that are usually initiated at the plasma membrane, become activated downstream of Sema4D[67,68]. Heterodimerization may also induce conformational changes in the intracellular C-terminal portion of Sema4D that differ from homodimerization and facilitate downstream signaling as has been reported for many receptor:co-receptor complexes[69]. Further work is required to delineate the pathways by which Sema4D-Sema7A reverse signaling directs morphological changes in neuronal and non-neuronal cells.

In addition to being a Satb2-target gene, Sema7A, appears to be a target of several early neuronal transcription factors such as NeuroD2, Tbr1, and Fezf2 (Fig. 1). Modulation of Sema7A levels downstream of these transcription factors, may provide a mechanism for regulating Sema4D activity not only during neuronal development but also in other biological situations such as in the neuro-immune axis where both of these Semaphorins have known roles[24,70,71]. For example, Sema4D expressed on microglia has recently been observed to be one of the primary mediators for interaction with plexin-containing astrocytes[72]. Similarly, elevated expression of Sema4D in neurons has been reported in association with Alzheimer's and Huntington's disease, where it appears to promote astrocyte reactivity[73].

The importance of the membrane localization of Sema4D for its biological function is highlighted by the SEMA4D-497P mutation described here in a patient with generalized tonic-clonic seizures. While SEMA4D-497P can homo- and hetero-dimerize and is predicted to form almost the identical protein structure, it is not correctly trafficked to the plasma membrane and shows evidence of incomplete glycosylation, yielding predominantly the immature (~120 kDa) version of the protein. It is worth noting that the 120 kDa and 150 kDa versions of Sema4D we observe here are both full length (containing the cytoplasmic domain), given that we detect both molecular weights when using a C-terminal tag (Fig. 4C). Although the existence of cleaved forms of Sema4D have been reported in other systems[58,74,75], we could not detect a C-terminal cleaved fragment despite our best efforts.

Glycosylation is a major post-translational modification involved in the sorting and trafficking of membrane and secreted proteins not only towards the plasma membrane but also between dendritic and axonal compartments[76,77]. Defects in glycosylation pathways are often associated with neurodevelopmental disorders including intellectual disability, epilepsy and neuropathies[78,79]. Incomplete glycosylation of the neurotrophin receptor TrkA, as is commonly observed in Hereditary sensory and autonomic neuropathy-associated TrkA variants[80], is associated with an immature, lower molecular weight protein that prematurely exits the cellular sorting system before reaching the plasma membrane and is thereby unable to activate the Ras-MAP kinase pathway[60,81]. Glycosylation can also alter protein function. For example, Sema5A has been reported to function as an attractant or as a repellent, depending on its glycosylation state[82]. To date, six O-linked glycosylation sites (T657, T663, S666, T670, T716, S722) have been observed on human Sema4D. These sites lie between the IG domain and transmembrane region, an area that is not included in the published crystal structure, so that their effect on protein structure and function cannot be accurately predicted.

Loss of membrane localization, and thus activity, of Sema4D upon mutation of residue 497, may lead to an epileptic state in multiple ways. We have shown that SEMA4D-497P interferes with callosal axonal outgrowth and the radial migration of cortical neurons (Fig. 7). Migrational defects in the neocortex are often reported in association with neurodevelopmental disorders, including epilepsy[1,2]. In contrast, aberrant axonal sprouting and neurite outgrowth are common responses to the damage induced by seizures[83]. Thus, modulation of cytoskeletal molecular pathways can influence epileptic endpoints and vice versa.

A direct effect of mutation at 497 may be interference with the previously reported function of Sema4D in receptor clustering during GABAergic synapse development[58,84], thereby leading to imbalances in connectivity and epilepsy. Accordingly, overexpression or infusion of the extracellular domain of Sema4D into the mouse hippocampus has been shown to both increase the formation of GABAergic synapses and

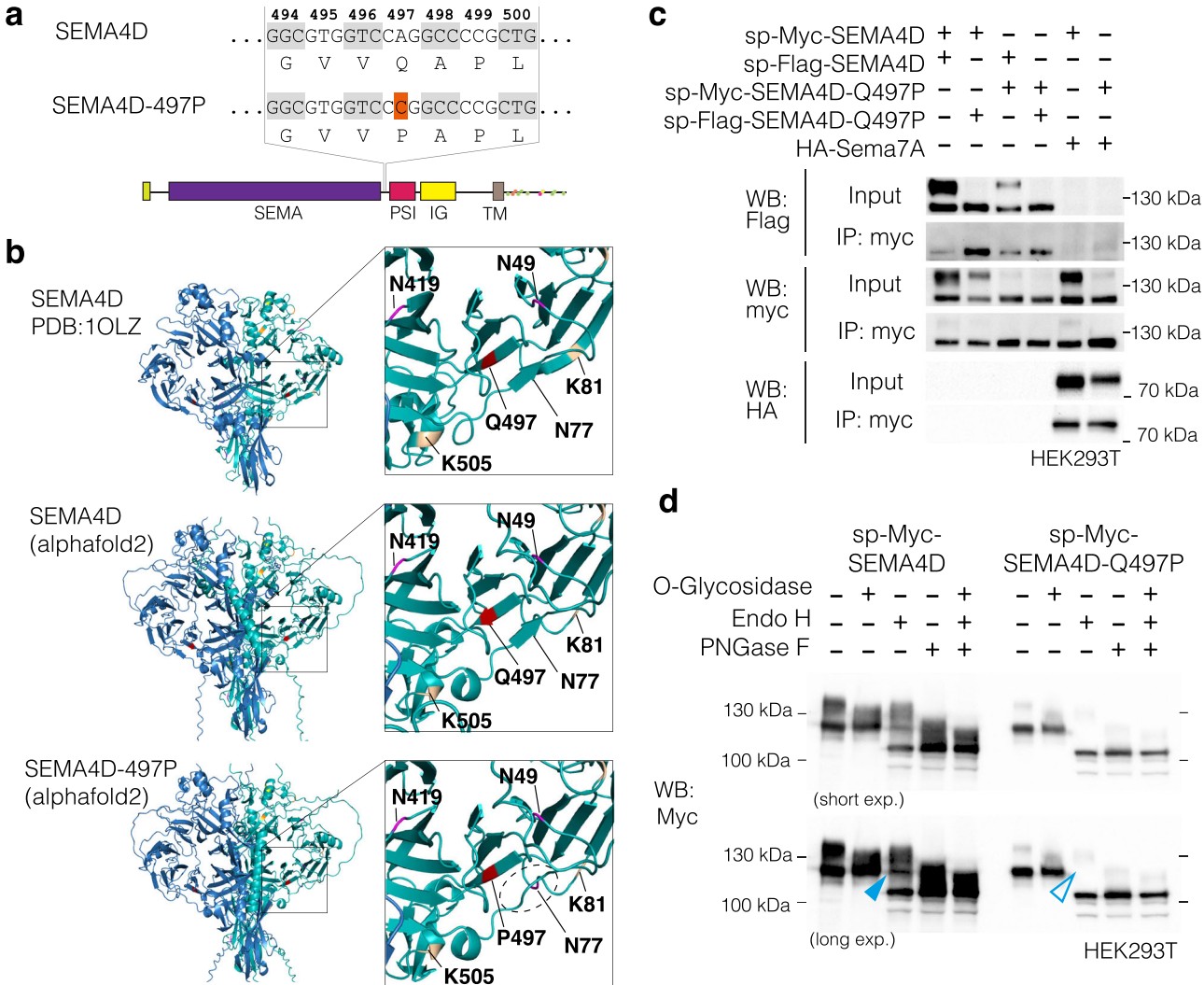

**Fig. 6 | Human Sema4D-497P Mutation retains homo and heterodimerization ability but is improperly processed. a** Schematic of adenosine to cytosine base mutation giving rise to a glutamine 'Q' to proline 'P' substitution at codon 497 between SEMA and PSI domains. **b** Ribbon diagrams and zooms of human SEMA4D solved crystal structure (PDB ID: 1OLZ, top) with alphafold2 predictions of wildtype SEMA4D (middle) and SEMA4D with 497P mutation (bottom). Codon 497 is highlighted red, glycosylation residues magenta, phosphorylation residues yellow, and ubiquitination residues ochre. A beta sheet parallel to codon 497 (dotted ellipse) is not predicted to form following 497P mutation. **c** Co-immunoprecipitation of signal peptide -myc-tagged human SEMA4D (sp-myc-SEMA4D) and the 497P variant (sp-myc-SEMA4D-Q497P) confirms that the mutant form can still form homodimers with sp-flag-hSema4D and heterodimers with mouse HA-Sema7A. While SEMA4D produces bands at ~150 kDa and ~120 kDa, the SEMA4D-Q497P variant predominantly generates the 120 kDa band. **d** De-glycosylation assay of sp-myc-SEMA4D and sp-myc-SEMA4D-Q497P using O-Glycosidase, Endo H and PNGase F. The same blot is shown at short and long exposures.

to suppress seizures in epileptic mouse models or in organotypic hippocampal epilepsy slice models[84–86]. Upregulation of Sema4D has also been found to be associated with high spiking regions, which presumably represent epileptogenic zones, as compared to paired low spiking regions in tissue surgically resected from focal cortical dysplasia patients[87], further linking dysregulation of Sema4D to epilepsy.

Similarly, elevated Sema7A expression has been observed in brain tissue of patients with temporal lobe epilepsy as well as in rat epileptic models. Downregulation of Sema7A was sufficient to suppress seizures, while upregulation had the opposite effect in the rat model[88]. Sema7A-deficient mouse cortices show fewer presynaptic puncta[89]. Based on our observation that Sema7A is a target of Satb2, the origin of epileptic symptoms in Satb2-Associated Syndrome (SAS)[15] may in part derive from a Sema4D:Sema7A complex deficiency.

In summary, the heterodimerization of Semaphorins with other Semaphorins—irrespective of their mechanism of membrane-association and their involvement in both forward and reverse signaling, gives rise to a multitude of different signaling and biological outcomes that still need to be explored and may have broad implications in both inflammatory neurodegeneration and neurodevelopmental disorders.

## Methods

### Animals

Mouse (*Mus Musculus*) lines used in this study were maintained in the animal facilities of the Charité University Hospital in Berlin, Germany. Mice were housed in 40-60% air humidity at 18-23 degrees Celsius with pellet food and water *ad libitum* on a 12 h light/dark cycle. All experiments included both sexes within litters without discrimination, and littermates of both sexes were randomly assigned to experimental groups during experimental procedures or tissue collection. Females were housed in groups of up to 5 animals per cage, while Males were housed singly after first exposure to female animals. *Satb2^fl/fl* Animals (*Satb2^tm2Rug*, MGI:6363495, Rudolf Grosschedl) were used for mosaic electroporations, or crossed to the *Nex^Cre* mouse (*Neurod6^tm1(cre)Kan*, MGI:2668659, Markus Schwab) for conditional knockout in the cortex.

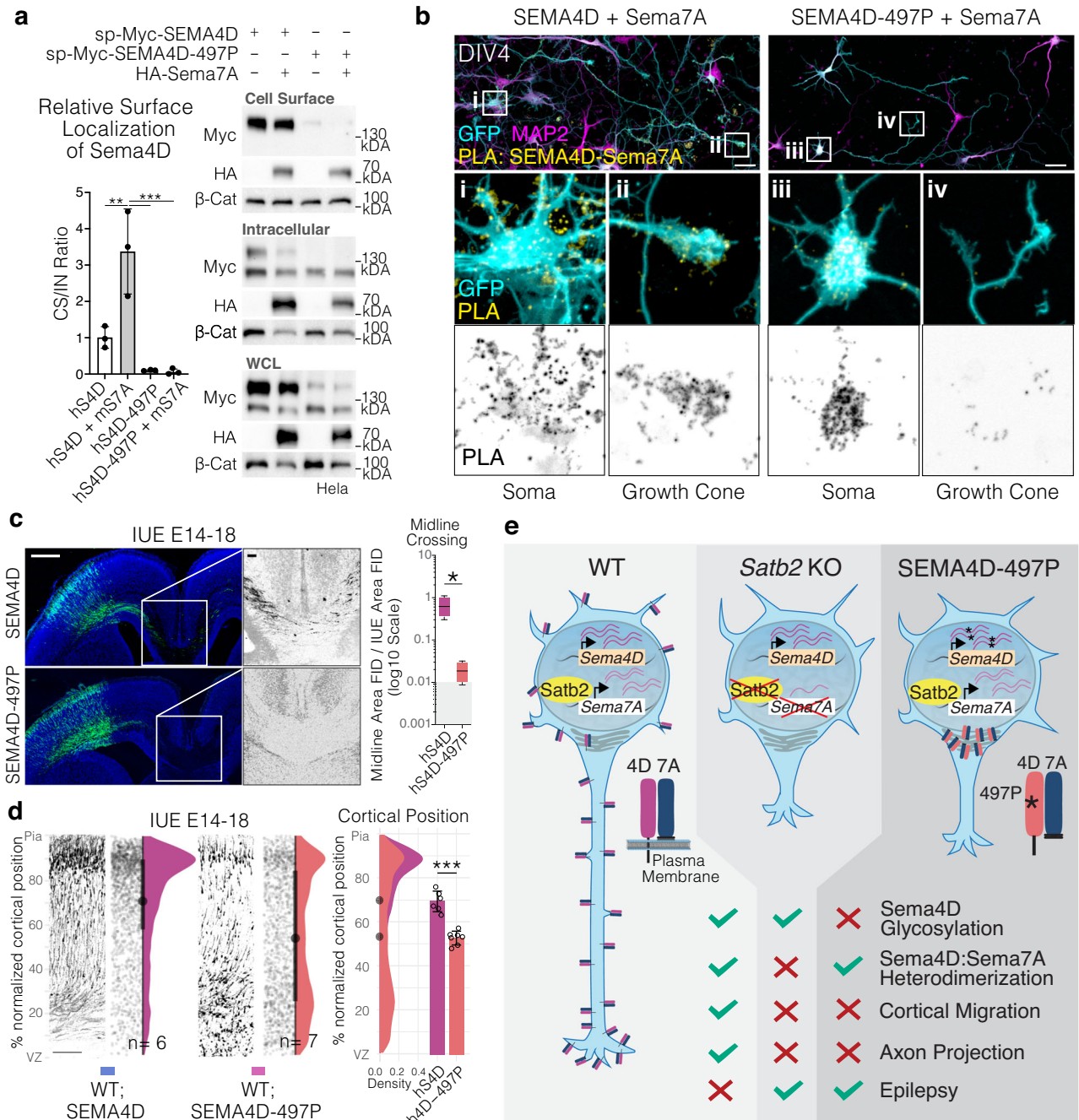

**Fig. 7 | 497 P Mutation abolishes localization of SEMA4D and Sema7A to the cell surface and growth cones and reduces neuronal migration and axon projection in vivo. a** Surface biotinylation followed by avidin pull down of human sp-myc-SEMA4D (abbr. hS4D) or human sp-myc-SEMA4D-497P (abbr. hS4D-497P) in the presence or absence of mouse HA-Sema7A (abbr. mS7A). Cell Surface to INtracellular (CS/IN) ratio of Sema4D was calculated by first normalizing each fraction to all Sema4D detected in whole cell lysate (WCL) where the complete calculation is CS/IN = (CS/WCL)/(IN/WCL). Plotted is mean+SD, using *n* = 3 separate experiments. Repeated measures ANOVA with Tukey's multiple comparison's. SEMA4D vs SEMA4D + Sema7A $p_{adjusted}$ = 0.0061, SEMA4D+Sema7A vs SEMA4D-497P $p_{adjusted}$ = 0.0008, SEMA4D + Sema7A vs SEMA4D-497P + Sema7A $p_{adjusted}$ = 0.0007. **b** Proximity Ligation Assay (PLA) detecting interaction between mouse HA-Sema7A and human sp-myc-SEMA4D or human sp-myc-Sema4D-497P. Scalebar = 10µm. **c** Midline panoramas and quantifications of human SEMA4D + GFP and SEMA4D-497P + GFP in utero electroporations. Two-tailed t-test with Welch's correction, SEMA4D vs SEMA4D-497P *p* = 0.0345. n_brains SEMA4D = 4, SEMA4D-497P = 4. Box and whisker plot whiskers as min-max with box bounds at

lower and upper quartiles and center line at the median. Scale bar in panoramas is 500 µm, while scale bar in midline magnification is 100 µm. **d** Neuronal migration profiles following in utero electroporation with SEMA4D or SEMA4D-497P, presented in the same format described in Fig. 1. Scalebar at bottom of SEMA4D IUE is 100µm. Representative electroporations are shown adjacent to raw data points and half violin plots of the total distribution of neurons across all brains with mean of means±SD cortical position on the right. Two-tailed *t* test, SEMA4D vs SEMA4D-497P *p* < 0.0001. n_brains SEMA4D *n* = 6, SEMA4D-497P *n* = 7. **e** Graphical model depicting general findings that Sema7A expression is regulated by Satb2, which promotes surface localization of Sema4D:Sema7A heterodimers (left). Upon deletion of Satb2 (middle), decreased Sema7A expression means heterodimerization with Sema4D is less able to occur, which coincides with deficits in axon extension and neuronal migration. Similarly, the Sema4D 497P mutation (right), which affects normal glycosylation and trafficking of Sema4D to the plasma membrane, results in deficits in neuronal migration and axon extension. For simplicity, panels denote *p* < 0.05 as *; *p* < 0.01 as **; and *p* < 0.001 as ***. All source data are provided in the Source Data file.

Wild-type, *Satb2fl/wt Nexwt/wt*, *Satb2fl/fl Nexwt/wt*, and *Satb2fl/wt NexCre/wt* were used interchangeably as wildtype controls. For ease of depiction and consistency, where relevant, controls have been labeled as *Satb2fl/wt NexCre*. The day of vaginal plug was considered embryonic day (E) 0.5, while the day of birth considers postnatal day (P) 0. Wild-type mice used were from an NMRI background. Mice were sacrificed by lethal injection of pentobarbital.

### Ethics approval
All mouse experiments were carried out in compliance with German law approved by the State Office for Health and Social Affairs, Council in Berlin, Landesamt für Gesundheit und Soziales (LaGeSo) under permissions G0079/11, G0206/16, G0184/20.

### Eukaryotic cell lines
HEK293T and Hela cell lines were obtained from DSMZ and maintained in DMEM (Life Technologies) supplemented with 10% FBS (Gibco), 1% Penicillin/Streptomycin and 1% GlutaMAX (Gibco). Cell lines were tested routinely for mycoplasma prior to experiments.

### Experimental procedures
For a list of antibodies, plasmids and key reagents please see Data S3. The Satb2 antibody used was self-generated previously[40]. Primers used are listed in Data S1.

### In utero electroporation (IUE) and culture of organotypic cortical slices
The procedure of IUE was performed as described[90] with minor modifications. Briefly, DNA plasmid vectors were injected into the lateral ventricle of mouse embryos at embryonic day 14 (E14) and electroporated brains were isolated and fixed at E18 with 4% Paraformaldehyde in PBS. Fixed brains were sectioned in 80 µm thick cryosections. Slice culture for live imaging was prepared according to published protocols with slight modifications[91]. Briefly, 250 µm thick cortical slices were sectioned in low melting media-agarose using a MICROM vibratome 24 h after IUE at E14.

### Cloning
Primers used for in situ probe templates and expression constructs are found in Data S1. Cloned expression constructs were deposited in addgene (plasmids #190643-190657).

Templates for in situ probes were amplified from mouse cortex cDNA using GoTaq polymerase (Promega) and ligated into the pGEM-T vector (Promega). Linearized probe templates were used to generate RNA probes by in vitro transcription using T7 or SP6 polymerase and dNTP DIG labeling mix (Roche).

Mouse Sema7A was amplified from mouse cortex cDNA using 'naked' primers in a first round of PCR with Q5-HF polymerase (NEB), whose product was used as a template in a second PCR using primers containing EcoRI and NotI restriction sites. EcoRI-Sema7A-NotI fragment was ligated into pAL2-T vector (Evrogen) yielding pGEMT-Sema7A. Domain deletion mutagenesis was performed on pAL2-T-Sema7A using the Q5 site-directed mutagenesis kit (NEB). For expression, full-length Sema7A or domain mutated Sema7A was ligated into EcoRI/NotI digested pCAG-eGFP (addgene# 164092), giving pCAG-Sema7A. After several failed attempts at an N terminal tag, an internal HA tag was designed after reviewing the crystal structure of Sema7A. Using pAL2-T-Sema7A, an HA tag was inserted using the Q5 mutagenesis kit at an exposed loop at amino acid 352, removing only 2 native amino acids to preserve the loop. This HA tagged form was then ligated into EcoRI linearized pCAG-eGFP to yield pCAG-HA-Sema7A.

As above, mouse Sema4D was amplified with Q5 polymerase using 'naked' primers, and a second round of PCR added NheI and NotI restriction sites to the amplicon, which was ligated into the pAL2-T vector (Evrogen) giving pAL2-T-NheI-Sema4D-NotI. After observing which amino acid begins the published structure of Sema4D and factoring where cleavage of the signal peptide (SP) is predicted to occur (SignalP-5.0), a myc tag was inserted prior to the phenylalanine at codon 23 using Q5 mutagenesis (NEB). Digested NheI-SP-Myc-Sema4D-NotI was then ligated into NheI/NotI digested pCAG-MCSAN, a pCAG vector with a modified MCS. pCAG-Sema4D-flag was cloned by amplifying from pAL2-T-Sema4D with primers containing flag sequence and also ligated into NheI/NotI digested pCAG-MCSAN.

Human myc- and flag-tagged versions of Sema4D (hSema4D) were designed similar to mouse sp-myc-Sema4D, where the tag is inserted after the predicted cleavage point of the signal peptide. Human Sema4D was amplified in two fragments 'A' and 'B' from a human cDNA library using GXL polymerase with the tag added via the primer to the first fragment. 'A' and 'B' fragments were inserted to EcoRI digested pCAG using NEBuilder to give pCAG-hSema4D. Similarly, the human Sema4D-Q497P point mutation was introduced by amplifying two fragments from human Sema4D and recombined into EcoRI opened pCAG- using NEBuilder (NEB).

### Chromatin Immunoprecipitation & qPCR
**Preparation of Chromatin.** E18 cortex was collected and snap frozen in liquid nitrogen and stored at -80 until use. ~75 mg of tissue was thawed by immediately placing in 1 ml of pre-cooled DMEM, and was homogenized by pipetting, and brought to a final volume of 2 ml. Tissue was then cross-linked by the addition of 200 µl of freshly made 10X crosslinking solution (16% formaldehyde, methanol-free (Thermo #28906), 1 mM EDTA, 0.5 mM EGTA, 100 mM NaCl, 50 mM HEPES-KOH pH 7.5) and rotated for 10 min. Fixation was stopped by addition of 250 µl 1.25 M glycine, rotating 5 min. Following a 5 min centrifugation (1350 g, 4 °C), tissue was resuspended in 10 ml ice-cold lysis buffer 1 (140 mM NaCl, 1 mM EDTA, 10% Glycerol, 0.5% NP-40, 0.25% Triton X-100, 10 mM Sodium Butyrate, 50 mM HEPES-KOH pH 7.55 with 1X Protease Inhibitor Cocktail (Roche) and rotated for 10 min at 4 °C. Following a 5 min centrifugation (1350 g, swing bucket rotor, 4 °C), cell pellet was resuspended in 10 ml lysis buffer 2 (cell lysis, 10 mM Tris-HCl, pH 8.0, 200 mM NaCl, 1 mM EDTA, 0.5 mM EGTA, 10 mM Sodium Butyrate with 1X Protease Inhibitor Cocktail) and rotated 10 min at 4 °C. 10ul was removed to count nuclei in a hemocytometer to confirm optimal concentration (aiming for 500 µl lysis buffer 3 for every $10^6$ nuclei). Following a 5-min centrifugation (1350 g, swing bucket rotor, 4 °C), nuclei were incubated for 10 min on ice in 100 µl lysis buffer 3 (10 mM Tris-HCl, pH 8.0, 100 mM NaCl, 1 mM EDTA, 0.5 mM EGTA, 0.1% Sodium Deoxycholate, 0.5% N-lauroyl sarcosine, 10 mM sodium Butyrate with 1X Protease inhibitor Cocktail) supplemented with 1% SDS. Prior to shearing, lysate was diluted in 1400 µl lysis buffer 3 not containing SDS. Chromatin was sonicated using a Bioruptor Plus (Diagenode) in v-bottomed Eppendorf tubes for 45 cycles using the high energy setting with a cycle of 30 seconds on 30 s off. 1/10 volume of 10% Triton X-100 was then added to chromatin and then spun at 4 °C prior to decanting soluble chromatin into a new tube. Chromatin was then stored at −80 for long-term storage. The resulting chromatin displayed a 200–600 bp smear of DNA following purification.

**Chromatin Immunoprecipitation.** Protein G Dynabeads (50 µl per IP, 25 µl pre-clear per IP = 75 µl per IP) were washed 3 times in PBS + 0.25% BSA. Chromatin was pre-cleared by incubating with 25 µl of washed beads and rotating 2hrs at 4 °C. At the same time, 12 µg Rabbit anti-Satb2 antibody (self-generated, against the peptide QQSQPTKESSP-PREEA) was incubated with 50 µl of washed beads per IP, rotating 2hrs at 4 °C. 10% of pre-cleared chromatin was then set aside as input. Excess buffer was removed from the antibody-bead mixture and incubated with the pre-cleared chromatin rotating overnight at 4 °C. Following the IP, beads were washed 6 times in RIPA (10 mM Tris-HCl, pH 8.0, 140 mM NaCl, 1 mM EDTA, 0.5 mM EGTA, 1% Triton X-100, 0.1% Sodium Deoxycholate, 0.1% SDS). Beads were then washed once in TE

buffer + 50 mM NaCl, and following a 3-min spin @ 960 g, all remaining supernatant was removed. 210 µl of elution buffer (50 mM Tris-HCl, pH 8.0, 10 mM EDTA, 1% SDS) was added to elute antibody-chromatin complexes from the beads at 65 °C for 15 min. The supernatant was then transferred to a new tube.

**DNA clean up.** Eluted chromatin and Input chromatin were processed simultaneously. Chromatin was mixed with 1/10 volume of 5 M NaCl and incubated overnight at 65 °C. 4 µl of RNase A was then added and incubated for 30 min at 37 °C. 4 µl of Proteinase K was then added and incubated for 30 min at 55 °C. Samples were then transferred to pre-spun phase lock tubes (5Prime, # 2302810) and an equal volume of Phenol:chloroform saturated with TE buffer (OmniPur®, Millipore #6805) was mixed by inversion prior to centrifugation at max speed. The aqueous phase was drawn off into a new epi and 1/10 volume of 3 M Sodium Acetate (pH 5.2), 2.5 volumes of 100% EtOH and Glycogen (to a final concentration of 0.05 µg/ul) were added. Tubes were vortexed well and DNA was allowed to precipitate overnight at -20 °C. DNA pellet was rinsed in ice cold 70% EtOH and dried until clear, then resuspended in 30ul ddH$_2$O.

**ChIP-qPCR.** The following primers were used for 150–200 bp fragment amplification within the selected region of Sema7A Intron1: F- CAGCCTAGTGTTGGGATGGT, R- ACAAGCAGGCTTGATTCCAT, and for the selected region of Sema7A 5′ Transcription Start Site (TSS): F- CGGGTAGCGAAGGTTTTCCT, R- CAGCCTTTTCTAGCTTTGCCG.

qPCR was performed using Sybr Green qRT-PCR Mastermix reagent. The efficiency of the primers was first tested using cDNA from cortex, measuring 5 serial dilutions (1:5) and analyzed on a StepOne Plus (Applied Biosystems) using the StepOne Software. The qPCR product was also run on a 1,5% Agarose gel to confirm amplicon size. qPCR was performed on 1 µl of 1:5 diluted cDNA from ChIP preparation, and 1:5 dilution of 1% Input.

Enrichment was calculated by first normalizing 1% of input used to 100% using equation (1), replicate 100% input Ct values (Ct$_I$) were then averaged to get a mean adjusted input (2). Enrichment was calculated as a percentage of input by calculating the ΔCt of IP'ed values compared to the adjusted input Ct (3).

Adjusted Input: $\forall$ Ct$_I$; (Ct$_I$ – 6.644) = Ct$_I$ adjusted

Mean adjusted Input = $\Sigma$(Ct$_I$ adjusted)/(n Ct$_I$ adjusted)

Fold Enrichment = 2^((Mean adjusted Input Ct) – (IP Ct))

## Culture of primary cortical neurons

For the analysis of axon specification in vitro, primary neuronal cultures were prepared as described before with minor modifications[92]. For PLA experiments, isolated neurons were nucleofected with the selected DNA plasmid according to the manufacturer's protocols (Mouse Neuron Nucleofector Kit, VPG-1001, and Amaxa 2b Nucleofection system, Lonza). For DIV2 and DIV4 measures of polarity in wildtype and Satb2−/− neurons, plasmid DNA was introduced by chemical transfection using Lipofectamine 2000 (Thermo Fischer) according to the manufacturer's protocol. Isolated neurons were transfected with the appropriate plasmids as above. Satb2$^{fl/wt}$ (for simplicity here labeled as WT), Satb2$^{fl/fl}$ (labeled as Satb2−/−) and Sema7A rescue neurons were cultured for 2 days and 4 days in vitro (DIV2, DIV4, respectively). Neurons were then cultivated in Neurobasal media (Gibco) supplemented with Glutamax, Penicillin– Streptomycin, and B27 at 37 °C in 5% CO$_2$. Neurons were fixed at the appropriate time using 4% paraformaldehyde (PFA) for 20 min at room temperature. For quantification of number of axons, cells were immunostained with markers for axons and dendrites and counted neurites were positive for Tau1 (1:1000, Millipore) and negative for MAP2 (1:1000, Novus). For unbiased measurements, we used a standardized pixel cross along with pre-defined criteria to measure the acute angle of the centrosome to the closest process at DIV2.

## De-glycosylation assay

Two wells of a six-well plate of HEK293T cells were transfected with hSema4D or hSema4D-Q497P using lipofectamine 2000 and cultured for 24 h. Each well was lysed in 100 µl Flag lysis buffer (50 mM Tris pH 7.4, 150 mM NaCl, 1 mM EDTA, 1% Triton X100) supplemented with 1X Protease Inhibitor Cocktail (Roche). For each enzymatic condition including undigested, 9 µl of lysate was incubated with Glycoprotein Denaturing Buffer (NEB) & heated for 10 min at 100 °C. Denatured lysates were then chilled on ice & briefly centrifuged before incubating with respective enzymes. For PNGase F lysates, reaction was brought to 20 µl by the addition of 2 µl Glycobuffer 2 (NEB), 2 µl 10% NP-40, 5 µl H$_2$O & 1 µl PNGase F (NEB). For O-Glycosidase lysates each reaction was brought to 20 µl by the addition of 2 µl Glycobuffer 2 (NEB), 2 µl 10% NP-40, 2 µl H$_2$O, 2 µl Neuramidase (NEB) & 2 µl O-Glycosidase (NEB). For Endo H lysates, each reaction was brought to 20 µl by addition of 2 µl GlycoBuffer 3 (NEB), 6 µl H$_2$O, 2 µl Endo H (NEB). Incubation with all three enzymes was performed in Glycobuffer 2. Reactions were incubated for 1 hr at 37 °C, and supplemented with 7ul of 4X Lammeli Buffer prior to boiling at 95 °C for 5 min for SDS-PAGE and western blot. Uncropped blots from all figures can be seen in Fig. S6.

## Immunofluorescence, in situ hybridization (ISH) and proximity ligation assay (PLA)

Immunofluorescence was performed in 2% BSA in PBS containing 1% Triton X-100 (Blocking solution). If Draq5 was to be used for nuclear stain, it was included in the blocking solution with secondary antibodies. In situ hybridization was performed according to[93]. For the Proximity Ligation Assay (PLA), cultured neurons were rinsed once in PBS containing 1 mM MgCl$_2$ and 0.1 mM CaCl$_2$ (PBS-MC) and fixed in PBS-MC containing 4% sucrose and 4% PFA for 20 min. PLA was performed according to the manufacturer's instructions (Sigma). Following amplification, coverslips with PLA neurons were incubated in a blocking solution for 30 min and then incubated with chicken anti-MAP2 (Novus) overnight in a blocking solution at 4 °C, followed by secondary antibody incubation and final mounting of coverslips for imaging.

## Microscopy and image processing

Imaging of fixed cortical sections was performed using an SL-1 confocal microscope (Leica). Specifically, after z stack acquisition, images were flattened to a maximum projection. Neuronal cell culture images and slice culture live imaging were carried out with a Spinning disc microscope (Zeiss).

## Co-immunoprecipitations and surface biotinylation assay

Twenty-four hours prior to isolation of protein, Hela cells were transfected (Lipofectamine 2000) with the appropriate DNA constructs. Cortices were dissected from NMRI embryos at E18.5 in cold PBS.

For lysis, cells and cortices were lysed in 200 µl ice-cold Flag lysis buffer (50 mM Tris pH 7.4, 150 mM NaCl, 1 mM EDTA, 1% Triton X100) per well or per embryo supplemented with either 1X Protease inhibitor cocktail (Roche) for the cell lines, or with 1X Protease inhibitor cocktail (Roche), 1x Phosstop (Roche), 5 µg/ml Pepstatin, 2.5 mM Na$_3$VO$_4$, 5 mM Benzamidine, 10 µg/ml Leupeptin, 1 mM β -glycerophosphate, 5 mM NaF, 5 µg/ml Aprotinin. Lysates were cleared by 10-15 min centrifugation at maximum speed at 4 °C on a tabletop centrifuge. The supernatant lysate was transferred to a new tube and protein concentration was measured by BCA. Equal amounts of protein were immunoprecipitated using the antibodies indicated in the figures for 2hrs at 4 C. Immunoprecipitates were washed 5x in lysis buffer before elution in 2.5x sample buffer and heating to 95 C for 4 min.

For Biotinylation, Hela cells transfected in a six-well plate were rinsed once in 0.5 ml ice-cold rinsing solution (0.1 mM CaCl$_2$, 1 mM MgCl$_2$ in PBS) and then incubated for 15 min with 0.5 ml Sulfo-NHS-SS biotin (1 mg/ml in rinsing solution) on ice. Cells were then rinsed briefly

in a quenching solution (Rinsing solution + 100 mM Glycine pH 2.5) on ice, then rinsed again 10 min in a quenching solution at 4 degrees to quench unbound biotin. Cells were rinsed once more in rinsing solution and then lysed in Flag lysis buffer supplemented with 1X Protease inhibitor cocktail (Roche), spun 10 min (max) at 4 °C, then supernatant transferred to a new tube and protein concentration measured by BCA. 50 µg of lysate was set aside as whole cell lysate, and 50 µg of lysate was incubated with 40 µl of avidin agarose beads (NeutrAvidin, Thermo #29201) for 2 h at 4 °C. Following a brief centrifugation, the supernatant (containing unbiotinylated intracellular fraction) was retained. Beads containing the biotinylated cell surface fraction were then washed 2x with Flag buffer and 2x with TBS, spinning down between washes at 500 g for 5 min on a tabletop centrifuge. Samples were then boiled at 95 °C for 5 min in lammeli buffer before running on SDS-PAGE.

## Statistics and reproducibility
Statistics were performed using GraphPad Prism 9.0. Values are supplied in the Source Data file and statistical details for all experiments are listed in the figure legends and in Data S2. Normally distributed data determined by Shapiro–Wilk and Kolmogorov–Smirnov tests were tested using two-tailed $t$ tests or one way ANOVA, and in cases where standard deviations were not equal the Brown-Forsythe ANOVA was used. Non-normally distributed data was analyzed using a Kruskal–Wallis test. Post-hoc analysis used either Tukey's multiple comparison across all conditions, Dunn's or Dunnett's when testing to a single control, or Dunnett's T3 when standard deviations were not equal. The exact test used is declared in the figure caption.

## Structural modeling
**Sema4D-Sema7A heterodimer superposition.** Protein structures of Semaphorin 4D (blue) and Semaphorin 7 A (yellow) were superimposed using Discovery Studio 5.0 to investigate the structural similarity. The crystal structures were retrieved from the RCSB Protein Data Bank (PDB) database[94], using the accession ID for Semaphorin 4D and Semaphorin 7 A - 3OL2 and 3NVQ, respectively. The alignment was based on specified residues. The best superimposition is determined based on a least-squares algorithm minimizing the interatomic distances of the equivalent atoms of the superimposed PDB structures. Only the $C^\alpha$ atoms were used since they show the least variation and are well determined[95]. The superimposition of both the protein structures showed a minor deviation (RMSD:0.64 Å) in the $C^\alpha$ atoms of the proteins.

**Prediction of human SEMA4D-Q497P structure using Alphafold2.** The colabfold implementation of alphafold2 using MMSeqs2 (https://colab.research.google.com/github/sokrypton/ColabFold/blob/main/AlphaFold2.ipynb)[59] was used to predict the folding of monomers of human SEMA4D and uman SEMA4D-497P using protein sequences without signal peptides and default parameters. With predicted monomers, the published crystal structure of Sema4D (1OLZ[55]) was used as a scaffold for dimerization using PyMOL[96]. Codon 497 and relevant post-translational sites (phosphosite.org[97]) were highlighted and images exported.

## Analysis of published datasets
Computation has been performed on the HPC for Research/Clinic cluster of the Berlin Institute of Health. Raw read data (fastq) from paired-end RNAseq experiments from P0 wiltdtype (SRR2027113, SRR2027115, SRR2027117) and Satb2−/− (SRR2027107, SRR2027109, SRR2027111) cortices was downloaded from GEO. Reads were aligned to mm10 using STAR and a count matrix generated using built-in --quantMode GeneCounts. RPKM for the Sema7A gene was calculated using the RPKM function in edgeR. For Satb2 ChIPseq, processed narrowPeak files from GSM2046905 were visualized in IGV browser.

## Exome sequencing
Informed consent was obtained prior to performing trio exome sequencing using a Sure Select Human All Exon 60 Mb V6 Kit (Agilent) for enrichment and sequenced on an Illumina NovaSeq6000 system (Illumina, San Diego, California, USA). Reads were aligned to the UCSC human reference assembly (hg19) with BWA v.0.5.8.1. Single-nucleotide variants (SNVs) and small insertions and deletions were detected with SAMtools v.0.1.7. Copy number variations (CNVs) were detected with ExomeDepth and Pindel. Variant prioritization was performed based on an autosomal recessive (MAF < 0.1%) and autosomal dominant (de novo variants, MAF < 0.01%) inheritance. This study was approved by the Ethics Committee of the Technical University of Munich, Munich, Germany (#5360/12S).

## Reporting summary
Further information on research design is available in the Nature Portfolio Reporting Summary linked to this article.

## Data availability
Published datasets analyzed are accessible from the Gene Expression Omnibus (GEO) database, www.ncbi.nlm.nih.gov/geo. Satb2-V5 ChIP-seq accession GSE77005, and Satb2KO RNAseq accession GSE68912. Exome sequencing data cannot be made publicly available due to data protection regulations. Source data are provided with this paper.

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

## Acknowledgements

This work was funded by Deutsche Forschungsgemeinschaft (DFG, German Research Foundation) [TA 303/14-1] (V.T.). P.B. was initially supported by a scholarship for research and innovation co-financed by Greece and the European Union under the National Strategic Reference Framework (NSRF/ΕΣΠΑ) and later by DFG research project grant # 66413881. A.G.N. was supported by DFG research project grant # 410579311 and a graduate scholarship from the Charité Univeristätsmedizin Berlin (Promotionsabschlussstipendium). E.K., Pr.Ba., and R.P. were supported by the DFG as part of the clinical research unit (CRU339), grant # 40952571. Sema4D-497P patch clamp experiments were supported by the Russian Science Foundation (RSF, grant 22-14-00232) (V.T.). We would like to thank Dr. Rudolph Grosschedl for providing the *Satb2^fl/fl* mouse, and Dr. Ingo Bormuth for IUE license applications and early technical support. We also thank Magda Krejczy and Prof. Dr. Britta Eickholt for the pCAG-FSF-GFP construct. Additionally, we would like to thank Dr. Mateusz Ambrozkiewicz for helpful scientific discussion. We also thank Roman Wunderlich, Ulrike Gunther and Daniel Richter for excellent technical support.

## Author contributions

P.B. and A.G.N carried out the bulk of the in vitro and in vivo experiments, molecular cloning, quantification, and analysis. P.B., A.G.N., M.R. and V.T. conceptualized the work. P.B. and A.G.N. assembled the figures and wrote the manuscript with input from the co-authors. M.R. and A.G.N.

edited the revised manuscript. K.Y. cloned in situ probes & provided technical expertise. R.D., D.L., E.R. and M.R assisted in performing biochemical assays. J.E. and T.B. characterized the Q497P mutation via exome sequencing. T.S. cloned human Sema4D and the 497P mutation, facilitated animal experiments & advised on molecular techniques. E.K., Pr.Ba. and R.P. performed structural modeling. P.D. assisted with revision experiments. K.T-T. provided technical expertise in Mass Spectrometry. M.R. and V.T. provided supervision.

## Funding

## Competing interests
The authors declare no competing interests.

## Additional information

[1]Institute of Cell Biology and Neurobiology, Charité-Universitätsmedizin Berlin, corporate member of Freie Universität Berlin, Humboldt-Universität zu Berlin, and Berlin Institute of Health, Charitéplatz 1, 10117 Berlin, Germany. [2]Department of Pediatrics, Dr. von Hauner Children's Hospital, University Hospital, Ludwig-Maximilians-Universität München, Munich, Germany. [3]Department of Pediatric Neurology and Developmental Medicine and Ludwig Maximilians University Center for Children with Medical Complexity, Dr. von Hauner Children's Hospital, Ludwig Maximilians University Hospital, Ludwig Maximilians University, Munich, Germany. [4]Institute of Human Genetics, Klinikum rechts der Isar, School of Medicine, Technical University of Munich, Munich, Germany. [5]Institute of Biochemistry, Charité -Universitätsmedizin Berlin, corporate member of Freie Universität Berlin, Humboldt-Universität zu Berlin, and Berlin Institute of Health, Philippstrasse 12, 10115 Berlin, Germany. [6]Core Facility – High-Throughput Mass Spectrometry, Charité – Universitätsmedizin Berlin, Corporate Member of Freie Universität Berlin and Humboldt-Universität zu Berlin, Core Facility – High-Throughput Mass Spectrometry, Am Charitéplatz 1, Berlin, Germany. [7]Institute of Physiology, Charité -Universitätsmedizin Berlin, corporate member of Freie Universität Berlin, Humboldt-Universität zu Berlin, and Berlin Institute of Health, Philippstrasse 12, 10115 Berlin, Germany. [8]Institute of Neuroscience, Lobachevsky University of Nizhny Novgorod, Nizhny Novgorod 603950, Russian Federation. [9]These authors contributed equally: Paraskevi Bessa, Andrew G. Newman. [10]These authors jointly supervised this work: Marta Rosário, Victor Tarabykin. ✉e-mail: victor.tarabykin@charite.de

