## [Peer Review File · Nature Communications]

Reviewers' Comments:

Reviewer #1:

Remarks to the Author:

In this study, Bessa and colleagues study mechanisms downstream of *Satb2* in the development and wiring of the cerebral cortex in mice. They find that the GPI-linked semaphorin *Sema7A* functions downstream of *Satb2* in processes such as neurite growth and neuron migration, the latter being dependent on effects on the centrosome localization. Interestingly, the effects of *Sema7A* appear to depend on another semaphorin, *Sema4D*, which binds *Sema7A* in cis at the same membrane and may act as a receptor. In the second part of this study, the authors investigate an epilepsy-associated mutation in *Sema4D* and show that it affects membrane localization and post-translational modification of *Sema4D* (and therefore of the *Sema7A*-*Sema4D* complex).

This is an interesting study that highlights a few novel mechanisms, such as the effect of *Sema7A* downstream of *Satb2*, the role of *Sema7A*-*Sema4D* interactions and the effects of post-translational modifications. However, currently it lacks a few essential controls (such as the effect of *Sema7A* overexpression in a WT background (Fig. 2)) and does not provide full insight into how *Sema7A* and *Sema4D* cooperate. Without the proper controls it is not possible to firmly establish that *Sema7A*/*Sema4D* function downstream of *Satb2* or whether their manipulation simply counteracts phenotypes induced by *Satb2* knockout. Further, the manuscript reads as two separate stories which could be connected better. Heteromeric interactions between different semaphorins have been described previously and semaphorins are known to act in cis. Further, semaphorins are known to be able to use other semaphorins as receptors. How exactly *Sema7A* and *Sema4D* cooperate remains unclear. Is *Sema7A* a cis ligand for the *Sema4D* receptor or is the function of *Sema7A* to regulate *Sema4D* cell surface expression? Or do both molecules form a receptor complex for a ligand presented in trans? And if so which ligand.

Other remarks

- p. 6, 7: The discussion about cell intrinsic and extrinsic mechanisms is rather confusing. *Satb2* functions cell intrinsically but may affect mechanisms in the cell and mechanisms that would require cues from the cellular environment. Perhaps the authors could clarify this point more.
- Fig. S1: I am a bit confused about the rescue experiment in S1B. It appears that restoring *Satb2* prevents axons from extending? Is this caused by more broad and perhaps ectopic expression or the overexpression?
- Fig. 1E: The ectopic expression of *Satb2* and *Sema7A* does not show a strong overlap. Are the sections shown consecutive? Which is the explanation for the incomplete overlap? Better illustration of these expression patterns is needed.
- Fig. 1F: *Sema7A* expression is difficult to observe. This panel would benefit from higher magnifications.
- Fig. 1G: It appears that *Sema7A* induces rescue in a subset of neurons that project axons dorsally in the corpus callosum.
- p. 11. It is stated that *Sema7A* re-expression restores certain phenotypes. The effects are actually more subtle and this should be emphasized as there is not full restoration. Also there is no effect on the longest neurite at DIV2. Incomplete rescue should also be emphasized for *Sema4D* KD in Fig. 5D-F.
- Fig. 4: PLA is used to show close proximity between *Sema7A* and *Sema4D*, but no evidence is shown for such interaction in the cortex in vivo. Can co-IPs be performed from cortical tissue?
- Several important controls are missing. For example, the effect of *Sema7A* expression in WT cortex (Fig. 2, see above) and effect of *Sema4D* KD in *Satb2* knockout (without *Sema7A* overexpression).

Reviewer #2:

Remarks to the Author:

In this manuscript, Dr. Bessa and Dr. Newman et al. have revealed that cortex SEMA7A expression is significantly reduced in Satb2 KO mice and it lead to inhibit nerve growth. Moreover, they have established that SEMA7A and SEMA4D associate in a transfection system, and that mutations in SEMA4D prevent its expression on the cell surface. Each of these discoveries is of scientific interest. However, I feel there lacks sufficient logical necessity linking these findings, and the entirety of the paper appears to be descriptive. This gives me the impression that several different works and a case report have been pieced together into a single work. Additionally, there are several errors that could have been avoided with a simple check, such as Typo and Legend notations. As it stands, I am currently not very positive about publishing this work. To make this work scientifically novel, the authors need to respond to the following comments.

1)

Most of the experiments showing the association between SEMA7A and SEMA4D are based on transfection systems, thus there lacks insight into whether this association is physiologically essential phenomenon. Co-localization finding at ISH displayed in Fig. 4C cannot support this sufficiently. The authors need to provide rigorous experimental evidence on the association of these two molecules under physiological circumstances, otherwise the binding of these molecules may be an artificial outcome of enforced expression.

2)

In Sup Fig3, the localization of SEMA7A and SEMA4D in the cortex is different in WT mice; why is the localization not similar as in Fig4C? Also, why is the expression of SEMA4D enhanced in Satb2 KO mice?

3)

The results from Fig3A, B, and C suggest that SEMA7A must be on the membrane and retain its integrin-binding domain. This makes sense that the formation of a dimer is crucial for SEMA7A; however, it remains uncertain why this result leads to the prediction that "it is complexed with other receptors" as pointed out in Lines 332-334. This is an important point that connects the first half of this paper with the second half, but I cannot catch the author's intention. Therefore, I would appreciate if additional explanation could be provided.

4)

If the Supplemental note only mentions a solitary case of epilepsy related to G497P SEMA4D, the graphic summary will be misleading representation of this mutation as a definitive cause of epilepsy. If there is no indication of family history of disease or GWAS data of multiple cases, it should remain speculative.

5)

In in vivo study on Fig7, do mice expressing this mutant SEMA4D develop epilepsy? Are

there any experimental data on survival or behavior abnormalities?

Fig3B : Show the figure with intact SEMA7A.

Fig3C : There is significant difference (*) in KCE, which is different from the description in Line 328-329.

Fig4 : The legend showing each Figure from A to E is wrong ordered.

Fig5 : The order of A→B→C would better to be changed to C→B→A, following Fig. 1 and Fig. 3, for consistence with the order of other Figures. In addition, GFP/Cre/Cre+SEMA7A tissue pics should be displayed in the same manner as the others.

Line 551 : Typo; it should be Fig6C.

Reviewer #3:

Remarks to the Author:

Bessa et al., report on a novel mechanism of how *Satb2* deficiency results in defective radial migration of neocortical neurons, and axon projections across the midline in the corpus callosum. The authors show that *Satb2* regulates expression of *Sema7a* mRNA in the developing neocortex, and *Sema7A* functions downstream of *Satb2* to control radial migration and axon pathfinding. For example, IUE of *Sema7a* in *Satb2* deficient neurons rescues in radial migration and axon outgrowth. Interestingly, *Sema7A* heterodimerizes with *Sema4D* in HEK293T cells; in primary neurons recombinant *Sema7A* and *Sema4D* are in close proximity, as assessed by proximity ligation, and *Sema4D* is required for *Sema7A* mediated radial migration and axonal outgrowth. This was demonstrated in *Satb2*(f/f) embryos electroporated with *Cre/GFP*, *Sema7a*, and *shSema4D* expression constructs. The *Sema4D* intracellular domain is necessary for cortical neuron radial migration, suggesting *Sema4D* functions as a receptor for *Sema7A*. A de novo human SEMA4D (497P) mutation, associated with severe seizures, was identified and assessed for defects in protein folding in silico. The mutation was predicted to cause steric hinderance of a nearby glycosylation site. Additional experiments revealed that SEMA4D (497P) still forms homo- and heterodimes when expressed in HEK293T cells, but exhibits defects in O-linked glycosylation and this abolishes cell surface localization. In neurons SEMA4D (497P), reduces neuronal migration and axon crossing across the midline in the corpus callosum.

Overall, this is a technically sound study, identifying *Sema7A* as a direct target of the transcription factor *Satb2*, and interestingly, provides evidence that *Sema7A* and *Sema4D* interact in cis, and that *Sema4D* function as a receptor for *Sema7A*. Moreover, a human SEMA4D mutation that blocks cell surface localization, impairs O-

glycosylation and is associated with severe seizures. While there are many strong elements to this study, there are several concerns that need to be addressed to support the main conclusions.

1. Experiments are largely based on overexpression of recombinant proteins, e.g. Sema7A and Sema4A in primary neurons or HEK293T cells. The study could be strengthened by showing expression and interaction of endogenous Sema7A and Sema4D.
2. Sema7A and Sema4D KO mice exist. Do these mice have defects in radial migration or midline crossing of axons in the corpus callosum?
3. An interesting finding is the identification of Sema4D as a novel receptor for Sema7A, in addition to PlexinC1 and integrin beta 1, however no evidence is shown that Sema7A and Sema4D function independently of PlexinC1 and integrin b1. The only evidence provided is the deletion of the integrin docking site. Additional evidence needs to be provided to claim that Sema7A association with Sema4D represents a novel receptor mechanism, rather than being an additional component of the PlexinC1 or integrin b1 receptor complexes described previously.

Minor concerns:

1. Please use conventional nomenclature for cell-autonomous versus non-cell autonomous functions of Satb2. It is not entirely clear what the authors mean with Satb2 deletion cell-intrinsically or cell extrinsic effects, this needs clarification.
2. Coronal brain sections were used to assess crossing of GFP+ upper layer neuron axons across the midline. In some of the sections the septum is visible while in others it is not. How did the authors ensure that midline crossing was absent (rather than axons at crossing at a different rostral-caudal level)?
3. In Figure 4, what is the molecular weight of recombinant Sema5A, how does it compare to the published literature?

Bessa, Newman et al., 2024 Revision

2024-04-22

Reviewer #1

We would like to thank the Reviewer for the positive and helpful assessment of our work. We have addressed all of the points raised.

(Remarks to the Author)

In this study, Bessa and colleagues study mechanisms downstream of Satb2 in the development and wiring of the cerebral cortex in mice. They find that the GPI-linked semaphorin Sema7A functions downstream of Satb2 in processes such as neurite growth and neuron migration, the latter being dependent on effects on the centrosome localization. Interestingly, the effects of Sema7A appear to depend on another semaphorin, Sema4D, which binds Sema7A in cis at the same membrane and may act as a receptor. In the second part of this study, the authors investigate an epilepsy-associated mutation in Sema4D and show that it affects membrane localization and post-translational modification of Sema4D (and therefore of the Sema7A-Sema4D complex).

This is an interesting study that highlights a few novel mechanisms, such as the effect of Sema7A downstream of Satb2, the role of Sema7A-Sema4D interactions and the effects of post-translational modifications. However, currently it lacks a few essential controls (such as the effect of Sema7A overexpression in a WT background (Fig. 2)) and does not provide full insight into how Sema7A and Sema4D cooperate. Without the proper controls it is not possible to firmly establish that Sema7A/Sema4D function downstream of Satb2 or whether their manipulation simply counteracts phenotypes induced by Satb2 knockout.

Further, the manuscript reads as two separate stories which could be connected better. Heteromeric interactions between different semaphorins have been described previously and semaphorins are known to act in cis.

Semaphorins have been previously shown to act as homodimers that bind their plexin receptors in trans. Several cases of reverse signalling have been shown, which we highlight in the discussion. However, to our knowledge this is the first instance of biological function of a Semaphorin heterodimer complex.

With regard to two disparate stories, we have taken note of the comments of the reviewer and have made changes throughout the manuscript and re-written the introduction and discussion to emphasize that for biological function in the neocortex you need membrane localization of Sema4D. This is dependent on both its interaction with Sema7A and correct glycosylation. Defects in either Sema7A expression/interaction or glycosylation results in disruption of cortical function that can be associated with neurodevelopmental diseases.

Further, semaphorins are known to be able to use other semaphorins as receptors. How exactly Sema7A and Sema4D cooperate remains unclear. Is Sema7A a cis ligand for the Sema4D receptor or is the function of Sema7A to regulate Sema4D cells surface expression? Or do both molecules form a receptor complex for a ligand presented in trans? And if so which ligand.

We are also intrigued by this, as this becomes an increasingly complex picture. Semaphorins have so far only been shown to act as homodimers. Semaphorin homodimers are then able to bind Plexins in

trans (Lu et al., 2021). The possibility that Semaphorins can heterodimerize, as demonstrated in our manuscript, adds another level signalling complexity to be further explored. It is possible that Semaphorin heterodimers are able to bind Plexins in trans to initiate signalling in neighboring cells. Our data, however, clearly indicates that it is the signalling initiated by Sema4D:Sema7A in the same cell, that is required for the radial migration and axon extension by that Sema4D:Sema7A expressing cell. We have provided an expanded exploration of these interactions in the discussion:

“ Sema4D and Sema7A directly interact to form a complex during neocortical development (Fig. 4). Our computational protein structure model suggests that Sema4D and Sema7A interact via the SEMA domain. This is concordant with previous studies that address the structure and function of this domain^{16,27,61}. The SEMA domain also mediates interaction of Semaphorins in trans to Plexins and neuropilins⁶². Heterotetramer complexes consisting of Semaphorin homodimers and dissociated Plexin monomers have been described^{54,63}. Similarly, Sema7A:Sema4D heterodimers may also be involved in forward signaling through interaction with other receptors on neighboring cells.”

Other remarks

- p. 6, 7: *The discussion about cell intrinsic and extrinsic mechanisms is rather confusing. Satb2 functions cell intrinsically but may affect mechanisms in the cell and mechanisms that would require cues from the cellular environment. Perhaps the authors could clarify this point more.*

- *Fig. S1: I am a bit confused about the rescue experiment in S1B. It appears that restoring Satb2 prevents axons from extending? Is this caused by more broad and perhaps ectopic expression or the overexpression?*

We thank the reviewer for making these points and have revised the text for explicit disambiguation of the cell extrinsic rescue and the cell intrinsic rescue in the introduction. Additionally, we have consistently remapped ‘cell intrinsic’ to ‘cell autonomous’ and ‘cell extrinsic’ to ‘non-cell autonomous’ to meet convention as per reviewer #3. We have modified the text as such:

“ ...Satb2 distinctly orchestrates both cell autonomous and non-cell autonomous transcriptional programs important for this developmental process.

In order to dissect whether Satb2 is required cell-autonomously for axon development of late-born projection neurons, we used a targeted *Satb2* mouse strain where exon 2 of *Satb2* is “floxed” (*Satb2^{fl/fl}*) and is then deleted by cre-mediated recombination. When *Satb2* is deleted from the developing dorsal neocortex using *Nes^{Cre}*, we observe that the corpus callosum is not formed and some axons reroute via the internal capsule (Fig. S1b), similar to what is observed in the constitutive mutant³³. Re-expression of *Satb2* in a selected number of neurons in the *Satb2*-deficient cortex is not sufficient to restore projections to the corpus callosum. This implies that there are non-cell autonomous (environmental) factors in a *Satb2*-deficient cortex that inhibit corpus callosum formation and confound possible understanding of the cell autonomous role of *Satb2* in axon projection. When *Satb2* is deleted in a mosaic fashion by only introducing Cre into only a few cells of the wild-type cortex, neuronal migration is perturbed, and axons do not project at all. Re-expression of *Satb2* in the same cre-expressing cell restores axon outgrowth in that cell demonstrating that axon outgrowth *per se* is controlled cell autonomously by *Satb2* (Figure S1c-e). “

- Fig. 1E: The ectopic expression of Satb2 and Sema7A does not show a strong overlap. Are the sections shown consecutive? Which is the explanation for the incomplete overlap? Better illustration of these expression patterns is needed.

The sections shown are consecutive. Satb2 expression is normally found in post-mitotic upper layer neurons of the cortical plate as shown previously (Britanova et al., 2008), which can also be seen in figure S1D. In figure 1e, we have overexpressed Satb2 in one hemisphere of the developing cortex and analysed expression of Sema7A in both electroporated and non-electroporated hemispheres by in situ hybridization. In the non-electroporated hemisphere we can observe the normal expression of Sema7A in post-mitotic neurons of the cortical plate, overlapping with the normal expression of Satb2. In the Satb2-electroporated hemisphere we can observe both the normal expression of Satb2 and Sema7A in the cortical plate, as well as in Satb2-overexpressing cells found in the subventricular zone. Thus, overexpression of Satb2 in vivo results in upregulation of Sema7A (Figure 1e). There is not 100% overlap possibly because additional factors found only in post-mitotic neurons are required for efficient Sema7A expression (such as NeuroD2 etc, highlighted in Figure 1c). Indeed, from recent publications we have also come to understand that Satb2 is now less considered a de facto transcription factor and more considered a chromatin looping factor (Wahl et al., 2024).

- Fig. 1F: Sema7A expression is difficult to observe. This panel would benefit from higher magnifications.

We do not show Sema7A expression in Figure 1F, this experiment shows that GFP and Cre expression overlap and demonstrate that Sema7A restores migration in the absence of Satb2 (presence of Cre, see figure S1d) (for migration also see Fig 1h).

- Fig. 1G: It appears that Sema7A induces rescue in a subset of neurons that project axons dorsally in the corpus callosum.

We have not observed rerouting of axons by Sema7A. Restoration of Sema7A expression in the absence of Satb2 results in extension of axons towards the corpus callosum in upper layer neurons.

- p. 11. It is stated that Sema7A re-expression restores certain phenotypes. The effects are actually more subtle and this should be emphasized as there is not full restoration. Also there is no effect on the longest neurite at DIV2. Incomplete rescue should also be emphasized for Sema4D KD in Fig. 5D-F.

We have moderated the language used in several instances:

We have moderated the language for Incomplete rescue of sema4D KD as follows: **“Only the construct encoding the full cDNA could rescue migration and moderately rescue axon outgrowth caused by Sema4D loss-of-function, while Sema4D-ΔIC was unable to repair these defects (Figure 5D-F, light and dark purple conditions).”**

We noted that there is no effect of Sema7A on the length of the longest neurite in Satb2-deficient neurons at DIV2, however the number of primary and end neurites are significantly different from the cre condition. We have added this sentence to make this more explicit: **“While the length of the longest neurite is unaffected at DIV2, Sema7A re-expression restored the number of primary and end neurites to WT levels”**

- Fig. 4: PLA is used to show close proximity between Sema7A and Sema4D, but no evidence is shown for such interaction in the cortex in vivo. Can co-IPs be performed from cortical tissue?

We thank the reviewer for this excellent suggestion. We now include endogenous co-immunoprecipitation data that demonstrate that Sema7A:4D complexes form in the neocortex during development. Immunoprecipitation of endogenous Sema4D from E18.5 neocortical lysates resulted in the strong co-immunoprecipitation of endogenous Sema7A (new Fig. 4f). This supports our three other investigations into this interaction, first, where we could observe a Sema7A and Sema4D complex in close proximity (<40nm; Alam et al. 2022) in neurons using PLA (Fig. 4F). Secondly, we show a series of co-immunoprecipitation of these proteins with multiple different tags in HEK293T cells to show that Sema7A and 4D can form a complex (Fig 4a, 4c).

- Several important controls are missing. For example, the effect of Sema7A expression in WT cortex (Fig. 2, see above) and effect of Sema4D KD in Satb2 knockout (without Sema7A overexpression).

We have performed the appropriate in utero electroporation controls of Sema7A overexpression in wildtype cortex and Sema4D knock-down in Satb2 knockout. They have been included in the new supplemental figure S4.

Reviewer #2

(Remarks to the Author):

In this manuscript, Dr. Bessa and Dr. Newman et al. have revealed that cortex SEMA7A expression is significantly reduced in Satb2 KO mice and it lead to inhibit nerve growth. Moreover, they have established that SEMA7A and SEMA4D associate in a transfection system, and that mutations in SEMA4D prevent its expression on the cell surface. Each of these discoveries is of scientific interest. However, I feel there lacks sufficient logical necessity linking these findings, and the entirety of the paper appears to be descriptive. This gives me the impression that several different works and a case report have been pieced together into a single work. Additionally, there are several errors that could have been avoided with a simple check, such as Typo and Legend notations. As it stands, I am currently not very positive about publishing this work. To make this work scientifically novel, the authors need to respond to the following comments.

We would like to thank the Reviewer for the helpful assessment of our work. We have addressed all of the points raised. With regard to disparate stories, we have taken note of the comments of the reviewer and have made changes throughout the manuscript and re-written the introduction and discussion to emphasize that for biological function in the neocortex you need membrane localization of Sema4D. This is dependent on both its interaction with Sema7A and correct glycosylation. Defects in either Sema7A expression/interaction or glycosylation results in disruption of cortical function that can be associated with neurodevelopmental diseases.

<Major concerns>

1)

Most of the experiments showing the association between SEMA7A and SEMA4D are based on transfection systems, thus there lacks insight into whether this association is physiologically essential phenomenon. Co-localization finding at ISH displayed in Fig. 4C cannot support this sufficiently. The authors need to provide rigorous experimental evidence on the association of these two molecules under physiological circumstances, otherwise the binding of these molecules may be an artificial outcome of enforced expression.

We thank the reviewer for this excellent suggestion. We now include endogenous co-immunoprecipitation data that demonstrate that Sema7A:4D complexes form in the neocortex during development. Immunoprecipitation of endogenous Sema4D from E18.5 neocortical lysates resulted in the strong co-immunoprecipitation of endogenous Sema7A (new Fig. 4f). This supports our three other investigations into this interaction, first, where we could observe a Sema7A and Sema4D complex in close proximity (<40nm; Alam et al. 2022) in neurons using PLA (Fig. 4F). Secondly we show a series of co-immunoprecipitation of these proteins with multiple different tags in HEK293T cells to show that Sema7A and 4D can form a complex (Fig 4a, 4c).

2)

In Sup Fig3, the localization of SEMA7A and SEMA4D in the cortex is different in WT mice; why is the localization not similar as in Fig4C? Also, why is the expression of SEMA4D enhanced in Satb2 KO mice?

We have observed that the expression of both Sema4D and Sema7A in neurons to dramatically change over corticogenesis, and that their overlap is maximal in the intermediate zone around E15-E16, which also coincides with upperlayer neuron axonogenesis. At E18.5 Sema7A expression starts to be restricted to somatosensory cortex and Sema4D expression becomes much more restricted to the midline (as seen in Supplemental figure 3).

We have repeated the Sema4D ISH at E15 and have replaced the panel in Figure 4 with new images (now figure 4B). We are unsure why Sema4D might be enhanced in Satb2 KO mice, it could be compensatory, or it could be due to multiple discordant transcriptional programs.

3)

The results from Fig3A, B, and C suggest that SEMA7A must be on the membrane and retain its integrin-binding domain. This makes sense that the formation of a dimer is crucial for SEMA7A; however, it remains uncertain why this result leads to the prediction that "it is complexed with other receptors" as pointed out in Lines 332-334. This is an important point that connects the first half of this paper with the second half, but I cannot catch the author's intention. Therefore, I would appreciate if additional explanation could be provided.

We appreciate the need for clarity here. Sema7a only has an extracellular domain and lacks an intracellular domain. Nevertheless, our results show that Sema7a expression can restore the function (migration/axonogenesis) of the Satb2-deficient cell in which it is expressed. To initiate signalling in the same cell in which it is expressed, Sema7a thus must be acting through another receptor that possesses an intracellular domain. We have amended the text in the results to clarify this point:

*"Since Sema7A lacks an intracellular domain and is reported to act as a ligand *in trans*⁴⁶, it was difficult to envisage how it mediated a cell autonomous rescue in Satb2 deficient neurons.... ...Another possibility is that Sema7A acts by complexing *in cis* with other transmembrane receptors capable of reverse signaling to effect cytoskeletal changes cell autonomously... ...Together these data show that Sema7A is acting as a membrane-associated receptor cell autonomously, and given its lack of an intracellular domain it was most likely in complex with other transmembrane receptors."*

4)

If the Supplemental note only mentions a solitary case of epilepsy related to G497P SEMA4D, the graphic summary will be misleading representation of this mutation as a definitive cause of epilepsy. If there is no indication of family history of disease or GWAS data of multiple cases, it should remain speculative.

While we are wary of overclaiming causality with regard to the Sema4D mutation, we found no other coding or splicing mutations in the patient and there is now substantial evidence in the literature linking dysregulation of Sema4D to epilepsy. Two independent studies show that increased expression of Sema4D (through either intra-hippocampal infusion of purified protein or virally-mediated overexpression) in different mouse model of epilepsy suppresses seizure activity (Acker et al. 2019, Adel et al. 2023). A third study uses acute treatment of an organotypic hippocampal slice epilepsy model with Sema4D to reach a similar conclusion, that Sema4D suppresses epileptiform activity by facilitating GABAergic synapse formation in the hippocampus (Kuzirian M. 2013). Additionally, comparison of surgically resected electrographically-graded paired tissue samples from focal cortical dysplasia (FCD) patients also revealed upregulation of SEMA4D mRNA in high spiking regions which presumably represent the epileptogenic zones, as compared to paired low spiking regions, (Srivastava A. 2021 Mol. Brain), further linking Sema4D to epilepsy.

We have included these references in the improved discussion.

5)

In vivo study on Fig7, do mice expressing this mutant SEMA4D develop epilepsy? Are there any

experimental data on survival or behavior abnormalities?

This would be interesting to examine. While it is possible for a mosaic genetic intervention to have effects on behaviour or epilepsy if enough cells are targeted, this would be outside the scope of the current study and would require further ethical approval. Our in-utero experiments are designed to limit overexpression to both prevent possible artefactual consequences of overexpression and to enable discernment of individual cells in the cortex at the specific window of neurogenesis and axonogenesis.

<Minor concerns>

Fig3B : Show the figure with intact SEMA7A.

We have added the overexpression panel of intact Sema7A from Figure 1 for comparison.

Fig3C : There is significant difference () in KCE, which is different from the description in Line 328-329.*

Thank you for noticing this mistake. This has been corrected in the text:

“...Mutation of this site interestingly disrupted the ability of Sema7A to promote axon outgrowth, and to a lesser extent migration, downstream of Satb2. “

Fig4 : The legend showing each Figure from A to E is wrong ordered.

We regret the error. This has been corrected.

Fig5 : The order of A→B→C would better to be changed to C→B→A, following Fig. 1 and Fig. 3, for consistence with the order of other Figures. In addition, GFP/Cre/Cre+SEMA7A tissue pics should be displayed in the same manner as the others.

We have converted GFP/Cre/Cre+SEMA7A tissue images in figure 1 to be black and white to make them consistent with how they are displayed in figures 3 and 5. For figure 5, we feel that the order A→B→C best serves the flow of the text and also serves the natural progression from figure 4 where we test the effect of simultaneous downregulation of Sema4D on neuronal migration in the Sema7A rescue condition.

Line 551 : Typo; it should be Fig6C.

We regret the error. This has been corrected.

Reviewer #3

We would like to thank the Reviewer for the positive and helpful assessment of our work. We have addressed all of the points raised.

(Remarks to the Author):

Bessa et al., report on a novel mechanism of how Satb2 deficiency results in defective radial migration of neocortical neurons, and axon projections across the midline in the corpus callosum. The authors show that Satb2 regulates expression of Sema7a mRNA in the developing neocortex, and Sema7A functions downstream of Satb2 to control radial migration and axon pathfinding. For example, IUE of Sema7a in Satb2 deficient neurons rescues in radial migration and axon outgrowth. Interestingly, Sema7A heterodimerizes with Sema4D in HEK293T cells; in primary neurons recombinant Sema7A and Sema4D are in close proximity, as assessed by proximity ligation, and Sema4D is required for Sema7A mediated radial migration and axonal outgrowth. This was demonstrated in Satb2(f/f) embryos electroporated with Cre/GFP, Sema7a, and shSema4D expression constructs. The Sema4D intracellular domain is necessary for cortical neuron radial migration, suggesting Sema4D functions as a receptor for Sema7A. A de novo human SEMA4D (497P) mutation, associated with severe seizures, was identified and assessed for defects in protein folding in silico. The mutation was predicted to cause steric hinderance of a nearby glycosylation site. Additional experiments revealed that SEMA4D (497P) still forms homo- and heterodimers when expressed in HEK293T cells, but exhibits defects in O-linked glycosylation and this abolishes cell surface localization. In neurons SEMA4D (497P), reduces neuronal migration and axon crossing across the midline in the corpus callosum.

Overall, this is a technically sound study, identifying Sema7A as a direct target of the transcription factor Satb2, and interestingly, provides evidence that Sema7A and Sema4D interact in cis, and that Sema4D function as a receptor for Sema7A. Moreover, a human SEMA4D mutation that blocks cell surface localization, impairs O-glycosylation and is associated with severe seizures. While there are many strong elements to this study, there are several concerns that need to be addressed to support the main conclusions.

1. Experiments are largely based on overexpression of recombinant proteins, e.g. Sema7A and Sema4A in primary neurons or HEK293T cells. The study could be strengthened by showing expression and interaction of endogenous Sema7A and Sema4D.

We thank the reviewer for this excellent suggestion. We now include endogenous co-immunoprecipitation data that demonstrate that Sema7A:4D complexes form in the neocortex during development. Immunoprecipitation of endogenous Sema4D from E18.5 neocortical lysates resulted in the strong co-immunoprecipitation of endogenous Sema7A (new Fig. 4f). This supports our three other investigations into this interaction, first, where we could observe a Sema7A and Sema4D complex in close proximity (<40nm; Alam et al. 2022) in neurons using PLA (Fig. 4F). Secondly, we show a series of co-immunoprecipitation of these proteins with multiple different tags in HEK293T cells to show that Sema7A and 4D can form a complex (Fig 4a, 4c).

2. Sema7A and Sema4D KO mice exist. Do these mice have defects in radial migration or midline crossing of axons in the corpus callosum?

Sema7A KO mice have been generated and used to study a myriad of different Sema7a functions including the assembly of synapses in the mouse olfactory bulb (Inoue 2018), the maturation of the barrel cortex (Carcea 2014), and various immunological responses (Kang 2011, Czopik 2006, Körner

2021). Radial migration and axon outgrowth in the neocortex have not yet been addressed in these mice. Interestingly however, *Sema7A*-deficient mice have been reported to exhibit abnormal axon outgrowth in the lateral olfactory tract (Pasterkamp 2003) and in regulating serotonergic innervation following spinal injury (Loy et al 2021), suggesting that it may be involved in promoting axon outgrowth along several different tracts.

Similarly, while *Sema 4D* KO mice have been used to address roles in immunological responses (Zhu 2016), platelet function (Zhu 2009) and tumorigenesis (Zhou et al., 2017) microglia function (Smith et al., 2015, Clark et al., 2021), and oligodendrogenesis (Taniguchi 2009), neuronal roles in the development of the neocortex have, to our knowledge, not yet been addressed.

3. An interesting finding is the identification of Sema4D as a novel receptor for Sema7A, in addition to PlexinC1 and integrin beta 1, however no evidence is shown that Sema7A and Sema4D function independently of PlexinC1 and integrin b1. The only evidence provided is the deletion of the integrin docking site. Additional evidence needs to be provided to claim that Sema7A association with Sema4D represents a novel receptor mechanism, rather than being an additional component of the PlexinC1 or integrin b1 receptor complexes described previously.

Semaphorins have been shown to homodimerize in cis, enabling their interaction with Plexins in trans to induce signalling in the plexin-expressing cell. We present the first demonstration of an heterodimeric Semaphorin family member complex, opening up the possibility of the existence of multiple combinatorial Semaphorin complexes. These heterodimeric complexes may or may not interact through Plexins or integrins in trans. We cannot however rule out the possibility that *Sema4D:Sema7A* heterodimers in addition to promoting signalling in the cells in which they are expressed (reverse signalling), also act as ligands by interacting in trans with plexins or integrins and inducing forward signalling on neighbouring Plexin-expressing cells. Given that our data is based on the mosaic, cell autonomous manipulation of *Sema4D* and *Sema7A* expression, it clearly demonstrates that it is the “reverse signalling” in the *Sema4D:7A* expressing cell that directs axon outgrowth and radial migration in these cells. We have clarified these points in the discussion:

“...*Sema7A* to be important for olfactory bulb, olfactory- and thalamo-cortical tracts, as well as for hippocampal and cortical axonal outgrowth and branching^{25,50,58}. However, these studies assumed that *Sema7A* acts only as a classical ligand, initiating signaling in a neighboring cell after binding in trans to one of its receptors, *Integrinβ1* or *PlexinC1*. The interpretation of these previous studies may need to be adjusted to account for the existence of Semaphorin:Semaphorin interactions and their ability to engage in reverse signaling as described here. In addition to the strong interaction of *Sema7A* with *Sema4D*, we also detected weaker binding between *Sema7A* and *Sema6A*, suggesting that other Semaphorin signaling combinations and reverse signaling cascades may exist. Until now, reverse signaling has only been demonstrated for a handful of transmembrane semaphorins acting alone through their own cytoplasmic domains; *Sema1a*⁵⁹, *Sema4A*³¹, *Sema4C*⁶⁰, *Sema6A*²⁸, *Sema6D*³⁰. Further experiments are needed to characterize the full extent of Semaphorin:Semaphorin interactions and their capacity to carry out reverse signaling.”

“...*Sema4D* and *Sema7A* directly interact to form a complex during neocortical development (Fig. 4). Our computational protein structure model suggests that *Sema4D* and *Sema7A* interact via the SEMA domain. This is concordant with previous studies that address the structure and function of this domain^{16,27,61}. The SEMA domain also mediates interaction of Semaphorins in trans to Plexins and neuropilins⁶². Heterotetramer complexes consisting of Semaphorin homodimers and dissociated Plexin monomers have been described^{54,63}. Similarly, *Sema7A:Sema4D* heterodimers may also be involved in forward signaling through interaction with other receptors on neighboring cells.”

Minor concerns:

1. Please use conventional nomenclature for cell-autonomous versus non-cell autonomous functions of Satb2. It is not entirely clear what the authors mean with Satb2 deletion cell-intrinsically or cell extrinsic effects, this needs clarification.

We appreciate this note. We have changed all previous mentions to follow the convention of cell autonomous or non-cell autonomous for clarity. Additionally, we have added this section to the introduction to further clarify the distinction in the Satb2 mutant cortex:

“...Satb2 distinctly orchestrates both cell autonomous and non-cell autonomous transcriptional programs important for this developmental process.

In order to dissect whether Satb2 is required cell-autonomously for axon development of late-born projection neurons, we used a targeted *Satb2* mouse strain where exon 2 of *Satb2* is “floxed” (*Satb2^{fl/fl}*) and is then deleted by cre-mediated recombination. When *Satb2* is deleted from the developing dorsal neocortex using *Nes^{Cre}*, we observe that the corpus callosum is not formed and some axons reroute via the internal capsule (Fig. S1b), similar to what is observed in the constitutive mutant³³. Re-expression of *Satb2* in a selected number of neurons in the *Satb2*-deficient cortex is not sufficient to restore projections to the corpus callosum. This implies that there are non-cell autonomous (environmental) factors in a *Satb2*-deficient cortex that inhibit corpus callosum formation and confound possible understanding of the cell autonomous role of *Satb2* in axon projection. When *Satb2* is deleted in a mosaic fashion by only introducing Cre into only a few cells of the wild-type cortex, neuronal migration is perturbed, and axons do not project at all. Re-expression of *Satb2* in the same cre-expressing cell restores axon outgrowth in that cell demonstrating that axon outgrowth *per se* is controlled cell autonomously by *Satb2* (Figure S1c-e). “

2. Coronal brain sections were used to assess crossing of GFP+ upper layer neuron axons across the midline. In some of the sections the septum is visible while in others it is not. How did the authors ensure that midline crossing was absent (rather than axons at crossing at a different rostral-caudal level)?

For panorama tissue images there are some cases where the septum is not visible, typically due to the nature of image acquisition and tiling. The most representative image from each electroporation was used. For quantification multiple sections from multiple electroporations were used corresponding to the same rostral-caudal level.

3. In Figure 4, what is the molecular weight of recombinant Sema5A, how does it compare to the published literature?

This is an astute observation and also something we noticed later and was additional motivation for us to clone Sema4D from scratch. Full length Sema5A should also be close to 130kDa. It is conceivable that full length Sema5A is also capable of binding to Sema7A. We have added this caveat to the text

“... We detected a strong binding affinity between Sema7A and Sema4D and a weak interaction with Sema6A, while the Sema5A construct did not generate a full length protein (Figure 4A).”

Reviewers' Comments:

Reviewer #1:

Remarks to the Author:

I thank the authors for their detailed response to my questions.

Regarding previous work reporting the ability of Semaphorins to form heterodimers or act as Semaphorin receptors, please see for example Sweeney et al. (2011, Neuron) and Rozbesky et al. (2019, Nature Comm). These previous studies do not diminish my enthusiasm for the current manuscript, but should be discussed in light of the current Sema7A-Sema4D findings.

In my version of the revised manuscript textual changes were not marked making it difficult to assess the various textual changes. However, I still feel that the two parts of the manuscript are not very well connected (i.e. at line 556). It perhaps would help to show the data on the role of Sema7A in Sema4D surface expression first and then transition into characterizing the mutation, glycosylation etc.

I do not agree with the reviewers that their data strongly support a reverse signaling role for a cis Sema7A (ligand) to Sema4D (receptor) interaction. The main conclusions from their data are that Sema7A functions downstream of Satb2 and that this Semaphorin regulates Sema4D cell surface expression. There is, however, insufficient evidence to suggest that Sema7A acts as a direct ligand for Sema4D in the cortex causing reverse signaling in the same cell. It is very well possible that other cis or trans interactors trigger or mediate the signaling. The inability of the Sema7A-KCE mutant to rescue the Satb2 phenotype seems to support this view. I would suggest to remove the reverse signaling statements from the manuscript and focus on the ability of Sema7A to regulate Sema4D cell surface expression.

Fig. 1E: In this panel the area of strongest expression of GFP and Satb2 (in the middle of the cortex) is not showing strongest Sema7A induction (this is more ventral: panel ii). Do the authors suggest that this discrepancy is caused by additional factors?

Fig. 1G: I apologize for this confusion. I intended to ask whether Sema7A specifically rescues a population of axons that runs in the dorsal part of the corpus callosum?

Reviewer #2:

Remarks to the Author:

The new Figures 4b and 4f clearly suggest the intrinsic association of SEMA7A and SEMA4D and provide data that solve my initial concerns.

In line with my comment 5), since there is only one case report, we must be cautious about directly linking SEMA4D mutation to epilepsy, as the authors noted. Therefore, I would say again that it is clinically important to determine whether mutant mice in the specific part of the cortex actually develop epilepsy. In my opinion, the authors have made a sincere effort on other points and the paper has been greatly improved, so I leave the decision on this point to the editor and look forward to seeing this point addressed in future research.

<Minor point>.

In new Figure 4f, the label "IP: SEMA4D" is obscured by the pasted image.

Reviewer #3:

Remarks to the Author:

In the revised and improved version of the manuscript, the authors have addressed some of the concerns raised by this reviewer. However, not assessing midline crossing in Sema7a KO or Sema4a KO mice, which already exist, to independently validate key findings in the current study is an omission.

* REVIEWERS' COMMENTS

Reviewer #1 (Remarks to the Author):

I thank the authors for their detailed response to my questions.

Regarding previous work reporting the ability of Semaphorins to form heterodimers or act as Semaphorin receptors, please see for example Sweeney et al. (2011, Neuron) and Rozbesky et al. (2019, Nature Comm). These previous studies do not diminish my enthusiasm for the current manuscript, but should be discussed in light of the current Sema7A-Sema4D findings.

We thank the reviewer for his/her work and commitment to improve our manuscript.

We have included the suggested references (Hernandez-Fleming 2017; Rozbesky 2019, Sweeney 2011) as follows:

“Members of Semaphorin Class 4 and Class 6 had recently been shown to be involved in reverse signaling during cell polarization and migration²⁸⁻³¹. In addition, the secreted Drosophila Sema-2a and 2b use the transmembrane Sema-1a to initiate signaling^{29,53,54}”

In my version of the revised manuscript textual changes were not marked making it difficult to assess the various textual changes. However, I still feel that the two parts of the manuscript are not very well connected (i.e. at line 556). It perhaps would help to show the data on the role of Sema7A in Sema4D surface expression first and then transition into characterizing the mutation, glycosylation etc.

We thank the reviewer for the suggestion. After careful consideration, we feel that the paper flows better when we first address the biological significance and molecular coordination of projection neurons before addressing pathological biochemistry.

I do not agree with the reviewers that their data strongly support a reverse signaling role for a cis Sema7A (ligand) to Sema4D (receptor) interaction. The main conclusions from their data are that Sema7A functions downstream of Satb2 and that this Semaphorin regulates Sema4D cell surface expression. There is, however, insufficient evidence to suggest that Sema7A acts as a direct ligand for Sema4D in the cortex causing reverse signaling in the same cell. It is very well possible that other cis or trans interactors trigger or mediate the signaling. The inability of the Sema7A-KCE mutant to rescue the Satb2 phenotype seems to support this view. I would suggest to remove the reverse signaling statements from the manuscript and focus on the ability of Sema7A to regulate Sema4D cell surface expression.

We believe that there may be a misunderstanding here. We do not claim that Sema7A is a ligand unlike the Drosophila Sema 2a/2b mentioned by the reviewer. Drosophila Sema 2a/2b are secreted proteins. This is not the case for Sema7A, which is associated with the plasma membrane through its GPI anchor. Our data suggests that Sema7A

instead functions as a co-receptor for Sema4D, both proteins forming a complex on the same membrane/cell. This is backed up by several lines of evidence:

- 1) We could not rescue axon extension or the migration of Satb2-deficient neurons with a soluble Sema7A (Δ MEM, Fig. 2)
- 2) We can detect interaction of both human and mouse Sema4D with murine 7A at the same plasma membrane in dissociated neurons by PLA (Fig. 4G-L; Fig. 7b)
- 3) Knockdown of Sema4D inhibits cell autonomously the function of Sema7A in Satb2-deficient neurons, indicating that it is functioning in the same cell as Sema7A (Fig. 5a-c)
- 4) Modeling of the interaction between Sema7A and Sema4D in cis yields only a very minor deviation of 0.64Å compared to a Sema7A homodimer (Fig. 4D).
- 5) Surface biotinylation assays in dissociated neurons indicate that co-expression of Sema7A and 4D enhances the plasma membrane localization of Sema 4D (Fig. 7A).

The Sema7A-KCE mutation alters the integrin recognition sequence, RGD. The inability of Sema7A-KCE to restore axon outgrowth in Satb2-deficient neurons in vivo, suggests that interaction with integrins is important for this process. It is unclear, however, whether the interaction of Sema7A with integrins is in trans, as previously reported (Pasterkamp et al., 2003). This is however likely, since at least at E18, Integrin beta 1 expression does not overlap with Sema7A, being predominantly expressed in the midline (Fig 1a). It may thus provide a guidance cue for axons that have already extended to the midline.

Fig. 1E: In this panel the area of strongest expression of GFP and Satb2 (in the middle of the cortex) is not showing strongest Sema7A induction (this is more ventral: panel ii). Do the authors suggest that this discrepancy is caused by additional factors?

We had previously addressed this issue in our revised text and included the following explanation:

In results:

„ChIPseq peaks³⁹ for transcription factors NeuroD2, Tbr1 and Fezf2 have also been found near the TSS of Sema7A (Figure 1C).“

In Discussion:

„In addition to being a Satb2-target gene, Sema7A, appears to be a target of several early neuronal transcription factors such as NeuroD2, Tbr1, and Fezf2 (Fig. 1). „

Additionally, in our first response to the reviewers we provided this explanation:

“In the Satb2-electroporated hemisphere we can observe both the normal expression of Satb2 and Sema7A in the cortical plate, as well as in Satb2-overexpressing cells found in the subventricular zone. Thus, overexpression of Satb2 in vivo results in upregulation of Sema7A (Figure 1e). There is not 100% overlap possibly because additional factors found only in post-mitotic neurons are required for efficient Sema7A expression (such as NeuroD2 etc, highlighted in Figure 1c). Indeed, from recent publications we have also come to understand that Satb2 is now less considered a de facto transcription factor and more considered a chromatin looping factor (Wahl et al., 2024).“

We have now also additionally added the following sentence in the article text in the

results:

“The less than 100% overlap of Satb2 overexpressing cells with Sema7A ectopic expression is likely because additional factors found only in post-mitotic neurons are required for efficient Sema7A expression, such as NeuroD2 (Figure 1C). Indeed, we have recently come to understand that Satb2 is less a *de facto* transcription factor, and more akin to a chromatin looping factor⁴².”

Fig. 1G: I apologize for this confusion. I intended to ask whether Sema7A specifically rescues a population of axons that runs in the dorsal part of the corpus callosum?

This is an interesting thought, that only a subset of upper neurons would be rescued by Sema7A. However, this is unlikely since Sema7A expression at E15 appears to be in all newly born neurons and our IUE was carried out at E14, and thus modifies all UL neurons but not the pioneering axons of Layer 5. We would also like to point out that the number of axons restored is proportional to the number of cells electroporated. In the examples shown we have a slightly larger electroporated population in the GFP control of Figure 1. This variation was controlled for in our quantifications.

Reviewer #2 (Remarks to the Author):

The new Figures 4b and 4f clearly suggest the intrinsic association of SEMA7A and SEMA4D and provide data that solve my initial concerns.

In line with my comment 5), since there is only one case report, we must be cautious about directly linking SEMA4D mutation to epilepsy, as the authors noted. Therefore, I would say again that it is clinically important to determine whether mutant mice in the specific part of the cortex actually develop epilepsy. In my opinion, the authors have made a sincere effort on other points and the paper has been greatly improved, so I leave the decision on this point to the editor and look forward to seeing this point addressed in future research.

We thank the reviewer for his/her work and commitment to improve our manuscript.

This is an interesting question that could be followed up in a subsequent paper. It would however require a different experimental paradigm (including much larger electroporated areas) and ethics approval.

<Minor point>.

In new Figure 4f, the label "IP: SEMA4D" is obscured by the pasted image.

We thank the reviewer. We believe this occurred during conversion to PDF.

Reviewer #3 (Remarks to the Author):

In the revised and improved version of the manuscript, the authors have addressed some of the concerns raised by this reviewer. However, not assessing midline crossing in *Sema7a* KO or *Sema4a* KO mice, which already exist, to independently validate key findings in the current study is an omission.

We thank the reviewer for his/her work and commitment to improve our manuscript.

It is not known whether *Sema7A* or *Sema 4D* KO mice have callosal axon defects. Unfortunately, we cannot acquire these animals, although we agree with the reviewer that this would be interesting for future studies. Our *in vivo* evidence using shRNA against *Sema7A* and *4D*, however, indicates that these factors are essential for axon outgrowth *in vivo*.